

# Three parameter based Streamflow elasticity model: Application to MOPEX basins in USA at Annual and Seasonal Scale

K. Goutam[1], A. K. Mishra[1]

[1] Glenn Department of Civil engineering, Clemson University, Clemson, USA

**Abstract**

We present a three parameter streamflow elasticity model as a function of precipitation, potential evaporation and change in storage applicable at both seasonal and annual scales. The model was applied

to 245 Model parameter estimation experiment (MOPEX) basins located in USA. We tested the performance of the modified equation and showed that at annual scale groundwater and surface water storage change also contributes significantly to the streamflow elasticity. It was found that there are significant differences among elasticities seasonally as well as spatially. Our observations indicated that streamflow elasticity of watersheds in eastern and western USA tends to vary in clusters and exhibit a

cyclic seasonal pattern. It was observed that watershed hydroclimatology has a major influence on seasonal streamflow elasticity due to precipitation in comparison to potential evapotranspiration and storage change.

## 1.  Introduction:

Several studies have emphasized on the sensitivity of streamflow to fluctuations in climate for river

basins across the world. Many studies used atmospheric forcings as an input to a hydrologic model to quantify the resulting changes in streamflow, for example, hydrologic models like Sacramento model (Nemec and Schaake, 1982), abcd model (Sankarasubramanian et al., 2001), SIMHYD and AWBM models (Chiew, 2006) were applied to evaluate the influence of various climate variables on streamflow. Another popular approach involves analytically deriving the sensitivity of streamflow

based on various conceptual models. For example, Schaake (1990) derived the streamflow sensitivity to precipitation using an empirical formula. Later, Dooge (1992) used a hortonian approach to quantify





sensitivity of runoff to climate change. Arora (2002) proposed the bivariate climate sensitivities based on potential evapotranspiration (PET) and precipitation (P). He derived the streamflow sensitivities to PET and P by using five different empirical formulae, namely the Schreiber equation (Schreiber, 1904), the Ol'dekop equation (Ol'dekop, 1911), the Budyko equation (Budyko, 1958), the Turc-Pike equation

(Pike, 1964; Turc, 1954), and the Zhang et al. equation (Zhang et al., 2001). More recently, Wang and Wang (2011) derived streamflow sensitivities to various climatic variables by combining an analytical solution of Budyko hypothesis and differential form of the Penman equation. Multivariate statistical methods are used to estimate the relationship between climate variables and streamflow at a particular site, for example, a bivariate linear regression method based on precipitation and temperature anomalies

(Potter et al., 2011) and a bivariate generalized linear regression method based on precipitation and potential evapotranspiration anomalies (Andréassian et al., 2015) were applied for determining the streamflow elasticities.

The sensitivity of the streamflow is usually expressed in terms of a non-dimensional quantity called as elasticity where a positive elasticity value indicates an increase in stream flow with increase in the

climate variable, whereas a negative elasticity value indicates a decrease in stream flow with an increase in the climate variable (Sankarasubramanian et al. 2001; Potter et al., 2011). All the elasticity-based models have shown that precipitation has a greater positive influence on streamflow. Fu et al., (2007) suggested that an increase in precipitation along with a positive deviation in temperature would result in lesser impact in streamflow, whereas a negative temperature deviation would result in a higher impact

in streamflow. Yang and Yang (2011) has identified that relative humidity has a positive influence, whereas net radiation and wind speed have a negative influence on streamflow. More recently, Andréassian et al., (2015) has identified a negative influence of potential evapotranspiration on streamflow.

Most of the elasticity models are applied at annual scales, however, the dominant control of climatic

and landscape properties on hydrologic responses are time scale dependent (Atkinson et al., 2002; Farmer et al., 2003; Wang and Alimohammadi, 2011). Therefore, there is an opportunity to investigate the elasticity of streamflow at the seasonal scale to explore the seasonal control of climate on water resource availability. Hence, a natural extension of the climate elasticity framework to a seasonal scale



would serve the purpose of understanding the climate and physical controls. Usually, most of the climate elasticity models assume that at annual scale both water storage change and groundwater loss are insignificant (Yang and Yang, 2011; Arora 2002). This assumption leads to a simplified water balance equation, which represents precipitation as a sum of evapotranspiration and streamflow.

Nevertheless, at a seasonal scale these changes cannot be neglected. In addition, this assumption was not adequately addressed in the studies related to annual climate elasticity models except in few studies (Wang, 2014). Few analytical studies (Chen et al., 2013; Istanbulluoglu et al., 2012; Jiang et al., 2015; Ye et al., 2015; Jiali et al., 2014) have utilised Darwinian concept put forth by Wang and Tang (2014) to model water balance changes that consider seasonality, storage change and extremes. However based

on elasticity model which takes into account both storage change and seasonality has not been investigated. Hence, this article mainly focuses on a) Testing the performance of elasticity model at annual scale by incorporating storage change as an influencing component and b) to evaluate the climate elasticity model at the seasonal scale.

The manuscript is organized as follows: in section 2) data sources and methodology were discussed.

Section 3) discusses the results by evaluating the modified climate elasticity model at an annual scale by incorporating precipitation, potential evapotranspiration and change in storage components. Further, we present the stream flow elasticity at a seasonal scale and evaluate their spatial variability. Finally, section 4) presents the conclusions along with the implications of these results.

## 2. Data sources and methodology:

### 2.1 Data

The hydrometeorological data (1948 to 2003) were collected from the Model parameter estimation experiment (MOPEX) basins located in USA, which are considered unaffected by human influence. In the MOPEX dataset, daily precipitation (P) is processed by the NWS Hydrology Laboratory, and streamflow (Q) is obtained from USGS National Water Information System (NWIS). Monthly actual

evaporation from 1986 to 2006 was obtained from the data set provided by Zhang et al., (2010) at 0.5° resolution hosted at University of Montana website. We selected Potential evapotranspiration (PET)





values from climate research unit (CRU) database available at 0.5° resolution based on an improved Penman-Monteith method (Harris et al., 2014). The actual evaporation and potential evaporation data are temporally averaged to annual scale values for annual analysis and to seasonal scale for seasonal values. Also, the gridded values of AET and PET are spatially averaged to the watershed scales for further analysis. This research is focused on the overlapped time period of all the data sets, i.e. 1983 to 2003. We considered 245 MOPEX basins located in US since there are no missing data for those regions.

### 2.2 Methodology:

**Modified Streamflow elasticity model applicable at seasonal and annual scales:**

Schaake (1990) first derived the relationship between elasticity of runoff (Q) to precipitation (P) according to an empirical formula as:

$$\varepsilon_{Q/P} = \frac{\Delta Q}{\Delta P} \frac{P}{Q} \qquad (1)$$

Later, Arora (2002) proposed the bivariate climate elasticity based on potential evapotranspiration (PET) and precipitation (P) as expressed in equation (2).

$$\frac{\Delta Q}{Q} = \varepsilon_{\frac{Q}{P}} \frac{\Delta P}{P} + \varepsilon_{\frac{Q}{PET}} \frac{\Delta PET}{PET} \qquad (2)$$

Where $\varepsilon_{\frac{Q}{P}}$ and $\varepsilon_{\frac{Q}{PET}}$ are elasticities due precipitation and potential evapotranspiration respectively.

Recently, Yang and Yang (2011) extended the bivariate framework (equation 2) by replacing the $\Delta PET$ term with the differential model developed by Roderick et al. (2007). This modification will eventually include precipitation (P), temperature (T), net radiation ($R_n$) and wind speed ($U_2$) leading to equation (3)

$$\frac{\Delta Q}{Q} = \varepsilon_{\frac{Q}{P}} \frac{\Delta P}{P} + \varepsilon_{\frac{Q}{T}} \frac{\Delta T}{T} + \varepsilon_{\frac{Q}{R_n}} \frac{\Delta R_n}{R_n} + \varepsilon_{\frac{Q}{U_2}} \frac{\Delta U_2}{U_2} \qquad (3)$$

We followed a similar approach as above studies but replaced the model-based results with observational data. Hence, to represent the co-variations of streamflow with precipitation, potential



evapotranspiration and storage change, we formulate a trivariate relationship as illustrated in equation (4)

$$\frac{Q_t - Q}{Q} = \varepsilon_{\frac{Q}{P}} \frac{P_t - P}{P} + \varepsilon_{\frac{Q}{PET}} \frac{PET_t - PET}{PET} + \varepsilon_{\frac{Q}{DS}} \frac{DS_t - DS}{DS} \qquad (4)$$

Where $Q_t$, $P_t$, $PET_t$ and $DS_t$ are stream flow, precipitation, potential evapotranspiration and storage

change. The water storage change averaged annually (seasonally) for year $t$ respectively. Whereas $Q$, $P$, $PET$ and $DS$ represent their long-term averages. Storage change ($DS_t$) is estimated as residuals of the water balance closure ($DS_t = P_t - AET_t - Q_t$). Figure 1 shows the MOPEX basins considered in our study and the spatial distribution of the long-term averages of $P$, $Q$, $PET$ and $DS$. The precipitation and streamflow are higher in the north-western and south-western regions of USA, whereas the potential

evapotranspiration is higher in the southern part of USA. The north-eastern and central parts witness a decrease in groundwater and surface storage, whereas in other regions it has increased. In situations where the underlying processes are unknown, it is possible to use a statistical model. Hence, a multivariate regression approach was adopted as in Andréassian et al., (2015). The streamflow elasticities ($\varepsilon_{\frac{Q}{P}}$, $\varepsilon_{\frac{Q}{PET}}$ and $\varepsilon_{\frac{Q}{DS}}$) were determined by fitting data on annual anomalies using a multiple

generalized least square (GLS) regression equation (Johnston, 1972) and model parameters are obtained by Maximum likelihood method. GLS can be used to perform linear regression when there is significant correlation between the explanatory variables used in the regression analysis. In these cases, ordinary least squares or weighted least squares can be statistically inefficient, or even give misleading inferences (Greene, 2008). Here GLS model was fitted to 245 selected watersheds with all

the values aggregated to annual means. Then, the significance of regression coefficient's were evaluated with a bootstrap approach as mentioned in Andréassian et al., (2015) by considering 1000 sample parameters with 95% significance level. We apply the equation (4) for calculating the seasonal elasticities. In this case, we replaced the annual mean with seasonal mean by aggregating the data into spring (March–May), summer (June–August), fall (September–November) and winter (December–

February) averages. Hence, we obtain statistically significant stream flow elasticities due to precipitation, potential evapotranspiration and storage change for each season.




**Evaluation of performance of the modified climate elasticity model at annual scale:**

We evaluated our trivariate elasticity model (Equation 4) against the bivariate elasticity regression model (equation 2) using Akaike information criterion (AIC) (Akiake, 1973) and Bayesian Information Criterion (BIC) (Schwarz, 1978).

AIC is given by equation as

$$AIC = -2\sum_{i=1}^{n}\log\left\{g(x_i \mid \theta_k\right\} + 2k \tag{5}$$

Where $n$ is the number of observations; $g(x)$ can be either equation (4) or equation (2); $\theta_k$ are the streamflow elasticities of the corresponding models and k is the number of parameters. Usually, the preferred model is the one in which the AIC value would be minimum. As evident from the equation (5), we can see that the first term in the equation tends to decrease with the model parameters, whereas the second term increases. Hence, AIC penalizes for the increase in number of parameters.

Bayesian Information Criterion (BIC) is computed using following equation:

$$BIC = -2\sum_{i=1}^{n}\log\left\{g(x_i \mid \theta_k\right\} + \log(n)k \tag{6}$$

As we can see that the BIC is similar to AIC except that the second term is multiplied by a factor of 0.5ln(n) with respect to AIC. As a result, BIC leans more towards less parameterized models. Overall, the preferred model would be the one which has both minimum AIC and BIC value.

## 3. Results and discussion:

The trivariate and bivariate models were applied to 245 watersheds across continental USA. In this section, we first discuss the validity of the three parameter elasticity model at an annual scale based on AIC and BIC criterion. Later, we assess how the inclusion of storage coefficient term has changed the stream flow elasticity of precipitation in trivariate model compared to previous elasticity model. Then, we evaluate the change in streamflow elasticity at seasonal scale and explore the various climatic controls on the computed elasticities.



### 3.1 Performance of the Modified climate elasticity model:

The proposed trivariate model was evaluated against the bivariate model using AIC and BIC. As mentioned earlier, AIC gives the relative quality of a statistical model with the preferred model having the lowest value irrespective of the sign. Figure 2, gives the AIC values of both the bivariate and

trivariate models for each of the 245 watersheds. We can see that in all of the watersheds, AIC values pertaining to the trivariate model is lesser than its counterpart bivariate model. To penalize the added storage coefficient term, we also computed the BIC for both the bivariate and trivariate models for each of the selected watersheds. Figure 3 illustrates the BIC values for these watersheds. This also indicates that BIC values for trivariate model is lesser than the bivariate model. These results indicate that the

proposed trivariate model is obviously a better choice when compared to bivariate model. The BIC result indicates that even if we give more weightage to penalize the added term, the modified model performs better than the bivariate method. In addition to that, figure 4 highlights the increase in the number of watersheds with the statistically significant stream flow elasticities due to PET in the trivariate equation. We can see that the number of statistically significant streamflow elasticity values

due to PET is lesser in number indicating that PET might be a less influencing factor to the streamflow variability at some of the selected MOPEX basins. Overall, this increase in number of statistically significant watersheds due to the addition of storage term further suggests that the trivariate model is a better fit than the bivariate model.

### 3.2 Changes in annual Streamflow elasticity using the trivariate model:

We calculated the difference in stream flow elasticity between the trivariate and bivariate model as $\varepsilon_{\frac{Q}{P_{Tri}}} - \varepsilon_{\frac{Q}{P_{Bi}}}$ . Figure 4 shows the spatial distribution (left side) of arithmetic differences between trivariate and bivariate streamflow elasticities due to precipitation and its probability distribution based on numerical values obtained from 245 watersheds shown as a violin plot (right side). The differences in

elasticity are mostly positive indicating that neglecting the effect of storage change has resulted in underestimating the elasticities due to precipitation in most of the watersheds. This underestimation



appears to be more significant in the western part of USA. In the central and north-eastern USA, the differences hover slightly above zero indicating not much change in elasticity in those regions. The violin plot also show that majority of the basins have underestimated elasticities due to precipitation. The changes in streamflow elasticities due to PET is not shown here because there is less overlap

between statistically significant watersheds of trivariate and bivariate equations.

### 3.3  Annual stream flow elasticities due to Storage change:

Figure 6 illustrates the spatial distribution (left side) of stream flow elasticity due to addition of storage change [or storage coefficient] whereas the violin plot (right side) shows the probability distribution of stream flow elasticities due to storage change. These values suggest that ground water has a net negative

effect on the streamflow indicating that at an annual scale, notable amount of water is being stored either as groundwater or as surface water. We can see that in the central and north-eastern part of USA, the elasticity values are in the range of 0 to -1. Nevertheless, as we go further west, we can see that the elasticity is decreasing below -1. It was observed that the high elasticity values are more prevalent in the western USA, which has also caused the significant difference in streamflow elasticity due to

precipitation. This indicates that in those regions the change in storage play an important role in streamflow variability at an annual scale and neglecting these changes would result in improper assessment of streamflow elasticities.

### 3.4  Spatial distribution of seasonal stream flow elasticities

**Stream flow elasticities due to precipitation:**

Figure 7 illustrates the seasonal (i.e., fall, spring, summer and winter) pattern of elasticities derived based on the precipitation. It is observed that, in fall season, higher elasticity values were observed for watersheds located in north-central and north-eastern USA, but as the season transforms to winter, these regions have lesser elasticity values compared to other regions. Whereas during spring some of the

watersheds located in north-central region seems to recover its lost elasticity. Finally, in summer the eastern part of USA also regains its elasticity.  We can see that the western part of USA does not follow



the same cycle as eastern region. There appears to be a lag in the response of streamflow to rainfall with the high elasticity values starting in winter in the western part of USA. However, it also appears to follow a cycle similar to what we have seen in the eastern part of USA. This clearly highlights the differential behavior of western and eastern USA streamflow elasticities due to precipitation.

**Seasonal stream flow elasticities due to potential evapotranspiration**:

Figure 7 highlights the seasonal pattern of streamflow elasticities due to evapotranspiration. It can be seen that there are comparatively less watersheds that have statistically significant elasticities due to potential evapotranspiration. Higher numbers of statistically significant watersheds were observed during spring followed by winter, summer and fall. We can see that during fall, the eastern region has a

negative elasticity indicating a decrease in stream flow due to increase in potential evapotranspiration. But, in the south western watersheds we can see a positive elasticity value indicating an increase in stream flow due to potential evapotranspiration**.** This increase can be viewed as an increase in available moisture locally causing more rainfall and subsequently more rainfall within the same season. This contrasting behaviour might be due to higher temperatures in southern USA increasing the potential

evapotranspiration and thus the capacity to withhold moisture. This might be similar to the precipitation recycling concept introduced by Eltahir and Bras [1998]. Similarly, in summer the central USA exhibits positive elasticity values whereas the north-eastern USA exhibits negative elasticity. During spring most of the watersheds exhibit negative elasticity and the higher magnitudes were observed for central part of US.

**Seasonal stream flow elasticities due to storage change:**

Figure 8 illustrates the seasonal pattern of streamflow elasticities due to storage change. It was observed that the seasonal elasticities exhibit change in spatial clusters. For example, the eastern USA seems to exhibit a cycle of negative elasticities in fall, and then its intensity decreased in winter, becomes almost

negligible in spring and exhibits positive elasticity in summer. However, the watersheds in south eastern coast seem to exhibit negative elasticities in summer followed by a decrease in negative elasticity values in fall and winter. This region exhibits positive elasticity values in spring whereas the rest of eastern USA exhibits positive elasticity in later season. Similarly, in case of western USA, it exhibits a cycle of



negative elasticity values starting in winter, followed by its decrease in spring, transforms into positive elasticity values during summer season, and again goes back to negative values in fall. Hence, we can see that the south eastern region has a seasonal cycle that leads the eastern region by a season, whereas the western USA lags behind the eastern region. The positive elasticity values indicate that for

particular season, there is an increase in water storage resulting in increase in runoff. This is mainly because, for that season, the streamflow is contributed by the storage sources in addition to precipitation. Similarly, the negative elasticity values for a particular season indicate that streamflow is withdrawn by the storage sources indicating an inverse relationship.

## 3.5  Relationship between elasticity and various hydroclimatic characteristics

We evaluated the relationship between elasticity variables with respect to the climate variables, that includes:  Precipitation (P), Streamflow (Q), storage change (DS), aridity index (AI) and evaporative index (EI). The objective is to find out which of the climatic variable may influence elasticity values. AI can be expressed as PET/ (P-DS), whereas EI is AET/ (P-DS). We estimated the non-linear relationship

using locally weighted scatterplot smoothing (LOWESS) with a smoothing parameter of 0.8. More information about LOWESS and its application in hydrology can be obtained from Lall (1995) and Cleveland, (1979).

**Streamflow elasticity due to precipitation:**

Figure 10 presents the relationship of streamflow elasticity due to rainfall with the hydroclimatic

variables (P, Q, DS, AI and EI). In the case of precipitation, the elasticity values in fall seem to be clustered around 2.5 to 5 mm rainfall range. However, there is no clear indication of how the rainfall elasticity changes with rainfall averages. In spring too, the elasticity values are clustered around 2 to 4 mm of rainfall range. However, it can be often seen that the watersheds witnessing low rainfall (<1 mm) has higher elasticity values in spring. The lower rainfall regions exhibit lower elasticity values in

contrast to what we have seen in spring, even though the elasticity values appear to be clustered around 2-4 mm average rainfall range. However, there is a poor relationship between winter season elasticities and its corresponding precipitation.





In the case of streamflow, the elasticities values in fall, spring and winter show an inverse nonlinear relationship with flow (Q). However for low flow values in summer, the elasticity values are also low, but at higher magnitude it follows an inverse exponential relationship as observed in other seasons. For Storage change, elasticity varies nonlinearly during spring and summer with storage change. During the fall and winter there does not appear to be any significant relationship with storage change. The precipitation elasticity increases in a nonlinear fashion with seasonal aridity index, specifically in spring and fall. However for higher values of elasticity they are decreasing in spring. During fall when the AI is within a range from 0 to 1.5, the elasticity increases sharply, but for higher values, the change seems to be more gradual. However, in summer there is a poor relationship between AI and elasticity of precipitation. The EI seems to have a nonlinear positive association with elasticities for all the seasons, however, a stronger association was observed in spring.

**Streamflow elasticity due to Potential evapotranspiration:**

The relationship between streamflow elasticity due to evapotranspiration and hydroclimatic variables are shown in Figure 11. In case of summer precipitation, the elasticities were clustered around rainfall of 3mm, whereas a similar type of cluster occurs in winter season for the rainfall below 5mm. During spring season there is a negative exponential association with precipitation. This indicates that in general, regions with more precipitation would result in less elasticity due to potential evapotranspiration. A similar observation can be made in the case of stream flow during fall, spring and winter seasons. However, in summer season, the elasticities are clustered for regions lesser than 1 mm streamflow. An increase in positive storage change during fall leads to a decrease in streamflow elasticity due to Potential evapotranspiration. In spring, there was no specific pattern was observed. In summer, the elasticities values are clustered in the range of -2 to 0 for storage change. Similarly, in winter, there are clustered in the range of 0 to 2 mm storage change. In the case of AI, there is a decrease followed by an increasing pattern was observed in fall season. In spring, there is a decrease in elasticity pattern with an increase in AI. In summer, the elasticity values are clustered around AI value of 1 followed by an increasing trend for higher AI values. During winter a decrease followed by an increase pattern was observed between AI and elasticity. The elasticity values in fall, spring and winter seems to decrease with increase in evaporative index, except for some watersheds with high evaporative





index. Only in summer, we can see that there appears a cluster at around 0.8 of EI, however there is an increase in elasticity pattern for higher EI values.

**Streamflow elasticity due to Storage change:**

The streamflow elasticities due to storage change appear to be more clustered around different rainfall range during fall, spring and winter season (Figure 12). However there is a decreasing trend between elasticities and summer precipitation. There is no definite pattern exists between elasticities and seasonal streamflow as most of these elasticities values were clustered at lower streamflow range. The elasticity changes in a decreasing fashion with mean storage change. This relationship does not show

any high clustering tendencies as in the case of precipitation and streamflow. Even with AI and EI, the storage change elasticity does not show any significant association, the elasticities are either clustered (AI) or there are scattered (EI). This suggests that this elasticity is not dependent much on the climatic factors; however, they are more dependent on physical characteristics of the basins (Ye et al., 2015).

## 4.   Conclusion:

The following conclusions are drawn from this study:

(a) The proposed three parameter streamflow elasticity model can be a better model than the two parameter elasticity model as it underestimated the stream flow elasticity due to precipitation. This is because the three parameter model was able to account for the covariation of precipitation, potential evaporation and storage change.

(b) It was observed that the seasonal elasticities vary spatially in two separate clusters (eastern and western USA). Both regions exhibit separate seasonal patterns which are cyclic in nature.

(c) The stream flow elasticities due to precipitation show a significant nonlinear association with the various climate variables, whereas, in the case of potential evapotranspiration and storage change the relationship seems weaker. We also observed that during spring the association is stronger than the

other seasons.



(d) It was observed that comparatively less number of watersheds have exhibited statistically significant elasticity values due to potential evapotranspiration (only 90 out of 245 basins at annual scale). Hence, evaluating elasticity due potential evapotranspiration requires special attention and additional research.

*Acknowledgement:* Authors are very much thankful to Dingbao Wang for his valuable discussion and sharing the data for preparing the manuscript. We are also thankful to Hongyi Li for his encouragement to submit our manuscript to this special issue.

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





# Figures:

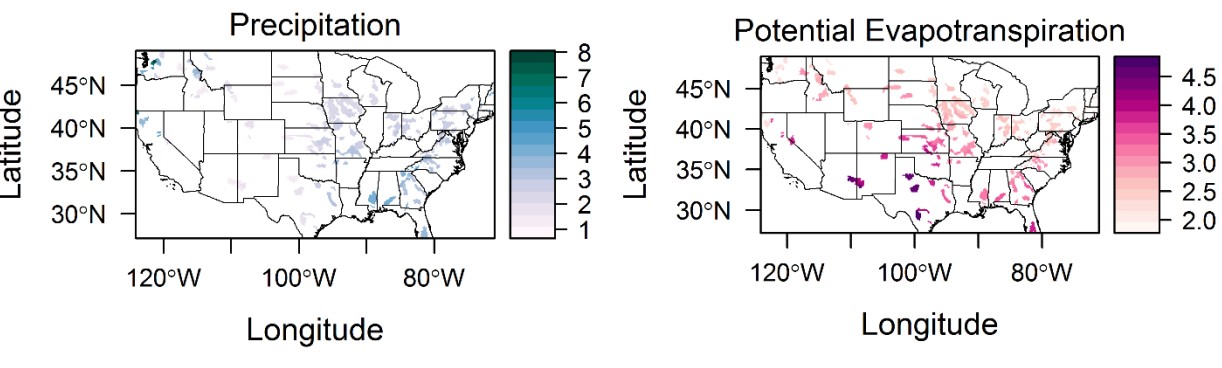

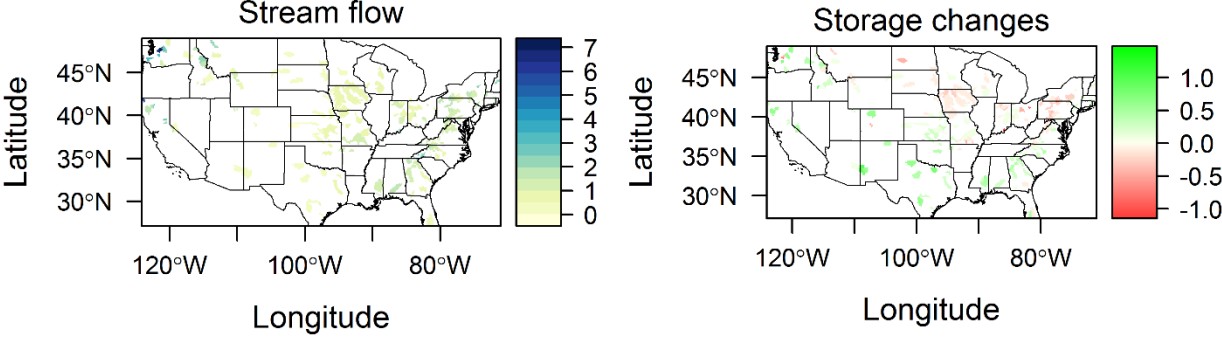

Figure 1: Long term annual averages of Precipitation (P), Potential evapotranspiration (PET), Stream flow (Q) and Storage change (DS) for the 245 MOPEX USA basins.





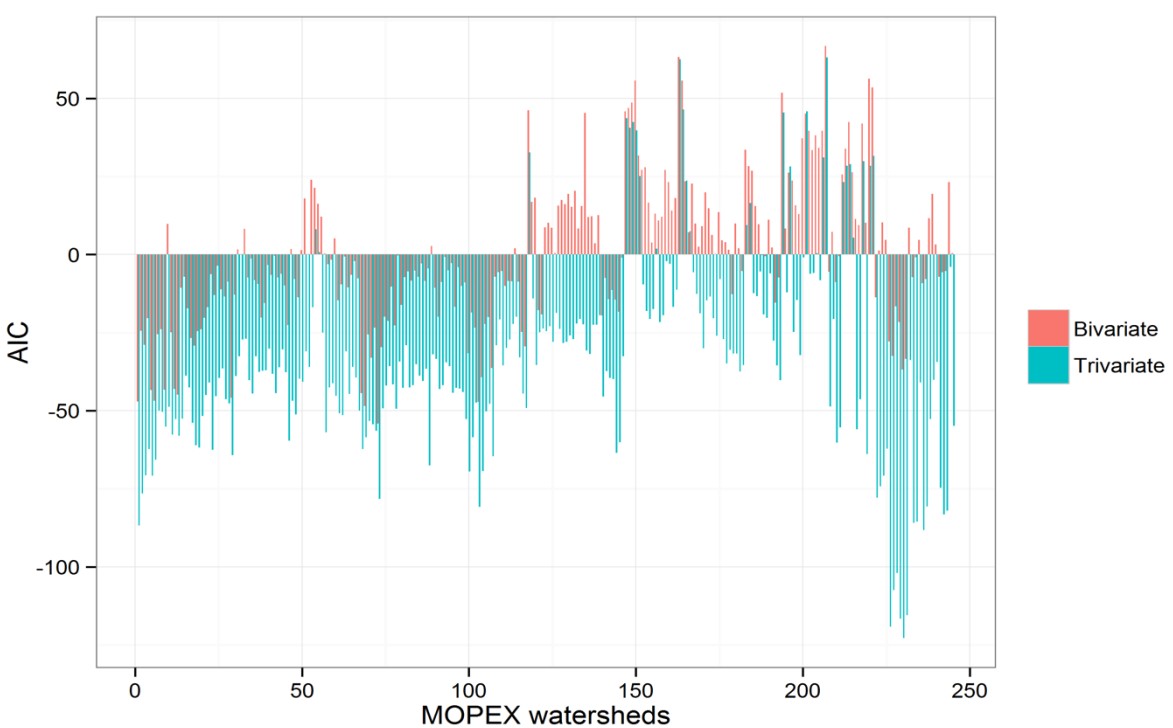

Figure 2: AIC values of the bivariate and trivariate models for the selected MOPEX basins.





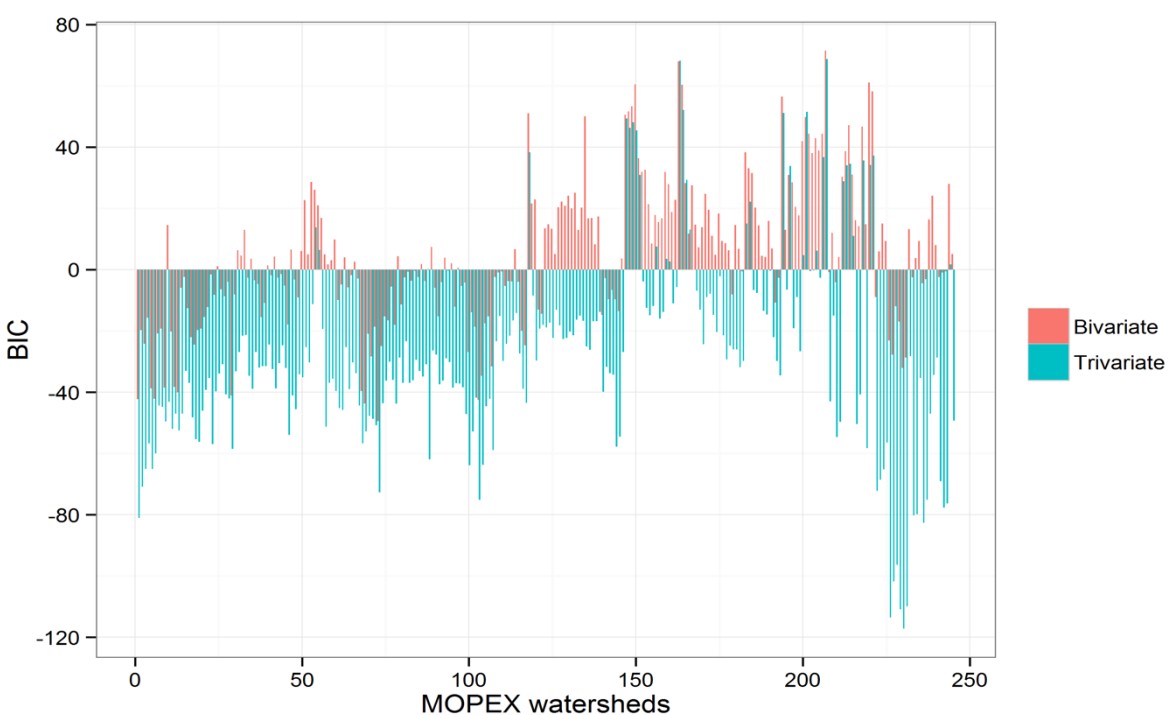

10   Figure 3: BIC values of the bivariate and trivariate models for the selected MOPEX basins.





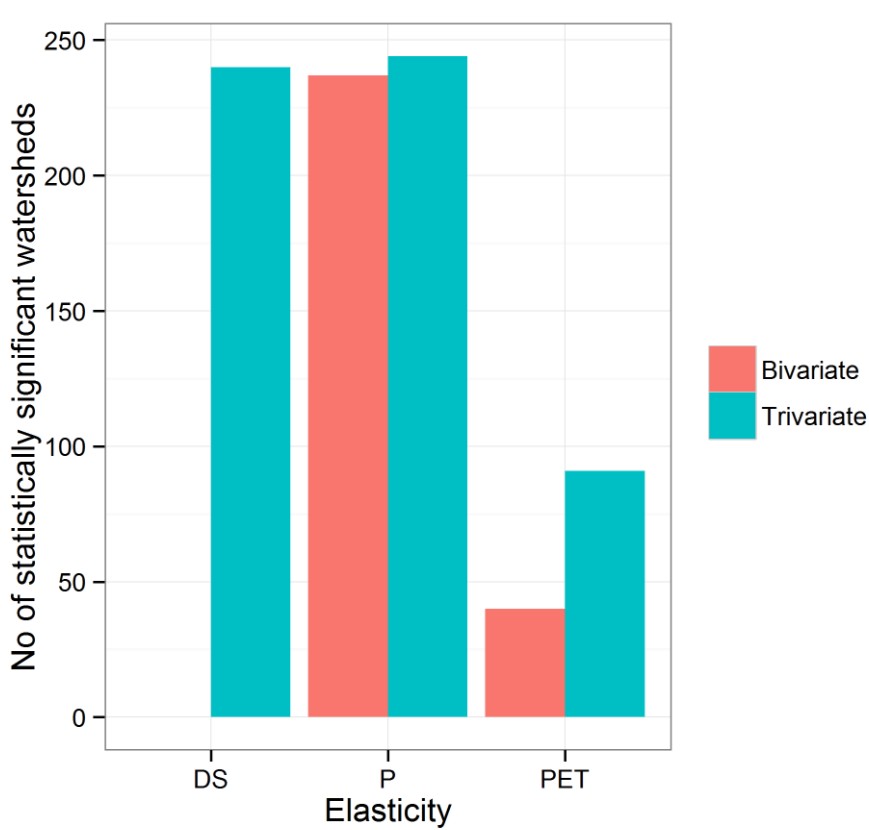

10     Figure 4: Number of statistically significant watersheds (p<0.1) for the elasticities of bivariate and trivariate equations.




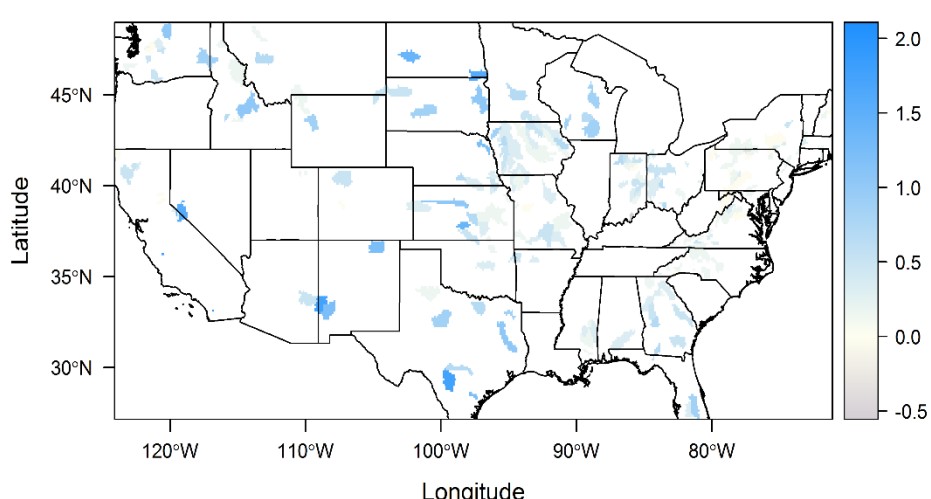
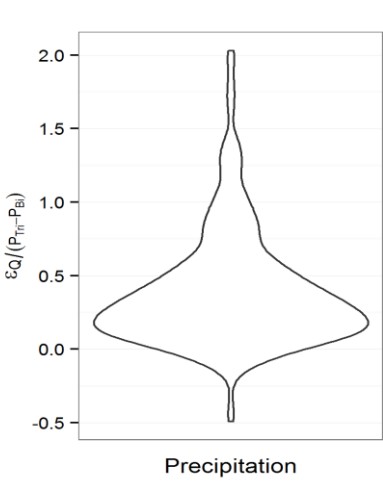

Figure 5: The difference in streamflow elasticity due to precipitation based on Trivariate and bivariate modelss. On the right side, a violin plot showing the distribution of elasticity differences.





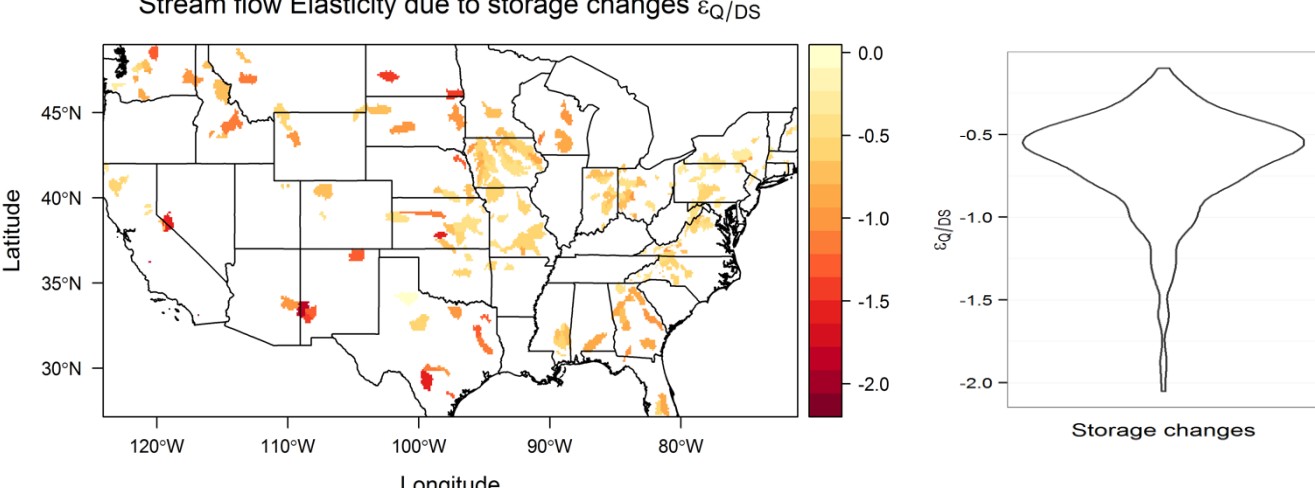

Figure 6: Annual Stream flow elasticity due to change in storage as derived from trivariate equation. On the left side, a violin plot showing the distribution of elasticity due to storage change.

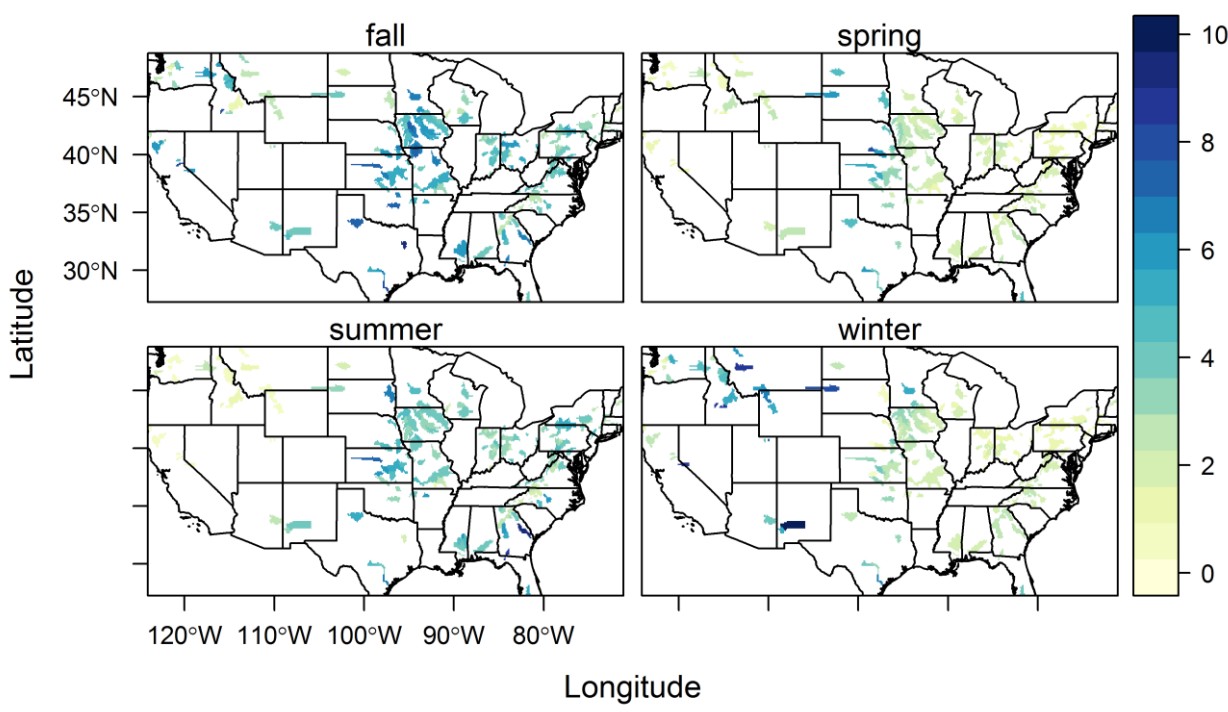

Figure 7: Seasonal distribution of Streamflow Elasticity due to precipitation



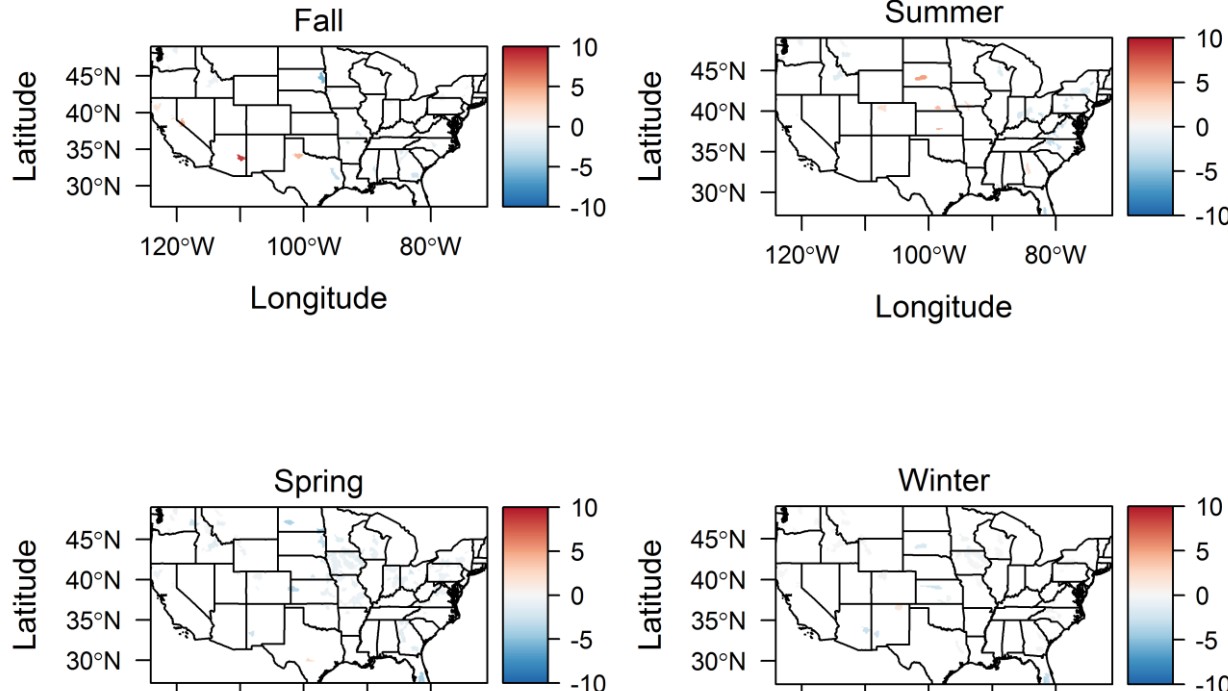

Figure 8: Seasonal distribution of streamflow Elasticity due to PET



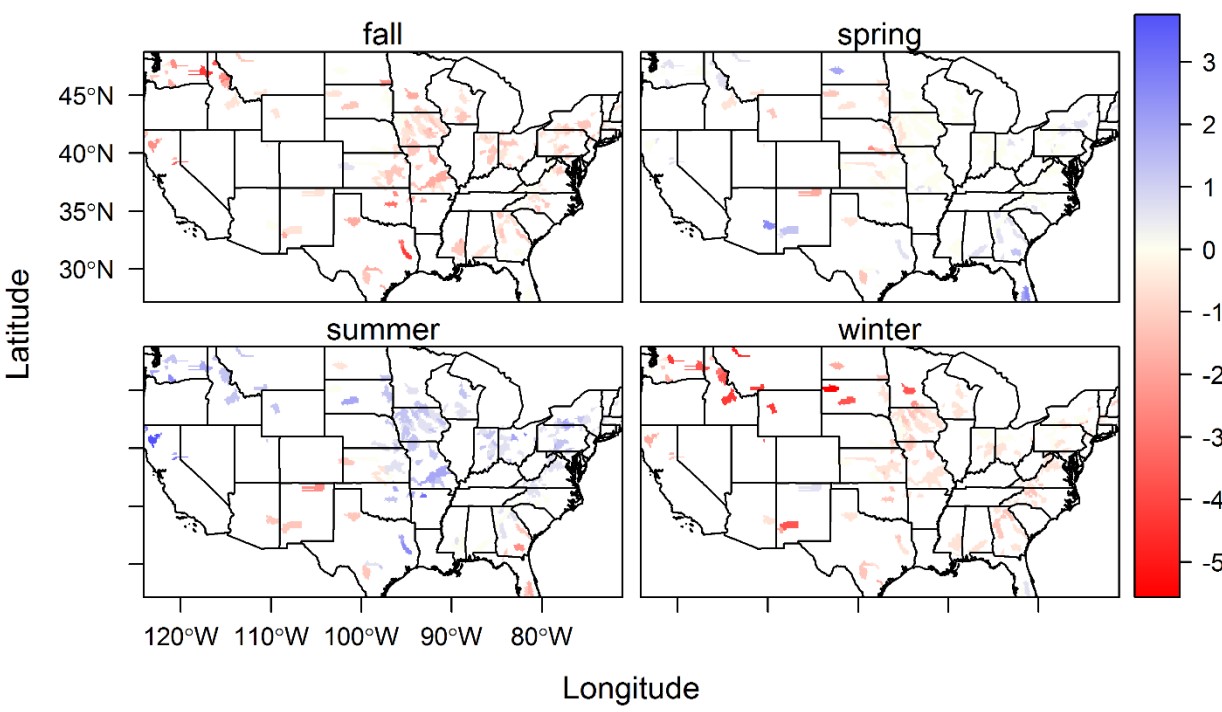

Figure 9: Seasonal distribution of Streamflow Elasticity due to storage change.







Figure 10: Impact of various climatic variables on the streamflow elasticity due to Precipitation





Figure 11: Impact of various climatic variables on the streamflow elasticity due to Potential Evapotranspiration





Figure 12: Impact of various climatic variables on the streamflow elasticity due to Storage change.