# Peer review of "Three parameter based Streamflow elasticity model: Application to MOPEX basins in USA at Annual and Seasonal Scale"

_Hydrology and Earth System Sciences, 2015_

## Short Comment (SC1) · 16 Feb 2016

Please note that this interactive comment is not intended as a full review of the paper. I just provide some comments that I think are worth considering and may strengthen the paper:

1. "Later, Dooge (1992) used a hortonian approach to quantify. . .' is a very vague description of what Dooge did. Essentially Dooge derived streamflow sensitivities based on different Budyko type equations (Ol'dekop, Schreiber, Budyko, etc etc). I think therefor this needs to be clarified compared to the current version of the text. He did something very similar to Arora (2002), but Dooge did not yet decomposed the role of climate into P and PET, and considered them as a lumped parameter (P/PET)
[Figure]

2. "All the elasticity-based models have shown that precipitation has a greater positive influence on streamflow" is not clearly formulated. Do you mean "All the elasticity-based models have shown that precipitation elasticity is positive"?

3. "Fu et al., (2007) suggested that an increase in precipitation along with a positive deviation in temperature would result in lesser impact in streamflow, whereas a negative temperature deviation would result in a higher impact in streamflow" is also unclear. If your formulate these influences in terms of elasticity values (see previous comment) your statement will get more clear.

4. "Yang and Yang (2011) has identified that relative humidity has a positive influence, whereas net radiation and wind speed have a negative influence on streamflow. More recently, Andréassian et al., (2015) has identified a negative influence of potential evapotranspiration on streamflow." Idem (see point 2&3)

5. "The hydrometeorological data (1948 to 2003) were collected from the Model parameter estimation experiment (MOPEX) basins located in USA, which are considered unaffected by human influence." What do you mean by "unaffected by human influence"; many of these catchments have land-surface conditions that are strongly affected by humans? I.e. catchment in the Midwest are mostly agricultural. Can you specify what you mean by "unaffected by human influences".

6. "Therefore, there is an opportunity to investigate the elasticity of streamflow at the seasonal scale to explore the seasonal control of climate on water resource availability." Please not that your concept of seasonal elasctity values is not per se novel. See e.g. Vano et al., 2015.

7. How does snow influence your study? Snow strongly affects your seasonal and annual water balances (Berghuijs et al., 2014, 2014). I suspect for example that snow links to your statement: "There appears to be a lag in the response of streamflow to rainfall with the high elasticity values starting in winter in the western part of USA. However, it also appears to follow a cycle similar to what we have seen in the eastern

part of USA. This clearly highlights the differential behavior of western and eastern USA streamflow elasticities due to precipitation."

8. One of the disadvantages of your approach is that the elasticity to water storage changes is derived from all residual (and uncertain) other data sources (Q, P, AET). Especially AET is uncertain as this cannot be directly measured. In the meanwhile there is also a way to calculate this metric using hydrograph recession analyses, see Berghuijs et al. (2016). Are you happy with your current approach or do you think that this method could actually make your results more robust?

9. I think the following statement is very specultive "We can see that during fall, the eastern region has a negative elasticity indicating a decrease in stream flow due to increase in potential evapotranspiration. But, in the south western watersheds we can see a positive elasticity value indicating an increase in stream flow due to potential evapotranspiration. This increase can be viewed as an increase in available moisture locally causing more rainfall and subsequently more rainfall within the same season. This contrasting behaviour might be due to higher temperatures in southern USA increasing the potential evapotranspiration and thus the capacity to withhold moisture. This might be similar to the precipitation recycling concept introduced by Eltahir and Bras [1998]." Do you have more evidence to support this?

10. Can you clarify your interpretation for: "Figure 8 illustrates the seasonal pattern of streamflow elasticities due to storage change. It was observed that the seasonal elasticities exhibit change in spatial clusters. For example, the eastern USA seems to exhibit a cycle of negative elasticities in fall, and then its intensity decreased in winter, becomes almost negligible in spring and exhibits positive elasticity in summer. However, the watersheds in south eastern coast seem to exhibit negative elasticities in summer followed by a decrease in negative elasticity values in fall and winter. This region exhibits positive elasticity values in spring whereas the rest of eastern USA exhibits positive elasticity in later season. S" Do you expect that this is the actual physical behavior of the catchment or can these seasonal changes also be induced

by the potential bias introduced in your result due to uncertaines in the components of water balances?

11. I do not think that the analysis of catchment properties influence on elasticity's is done rigorously. Can you make this part of the analysis a bit more appealing and convincing? Also, why did you choose these catchment properties?

12. The discussion part of this paper is somewhat thin in my opinion.

13. Figure 1 & 5: can you please be accurate in what we are looking at, and specify the units (even if they're dimensionless).

14. Figure 6: On the left side? Or on right side?

15. Figure 8: this colorbar makes it impossible to read the figure well

16. Figure 10-12: what do these blue thin lines resemble?

17. Are the following studies relevant to discuss or acknowledge: Renner et al, 2012, Sivapalan et al, 2011; Harman et al., 2011; Xu et al., 2013?

References - Berghuijs, W. R., Woods, R. A., & Hrachowitz, M. (2014). A precipitation shift from snow towards rain leads to a decrease in streamflow. Nat. Clim. Change, 4(7), 583-586.

- Berghuijs, W. R., M. Sivapalan, R. A. Woods, and H. H. G. Savenije (2014), Patterns of similarity of seasonal water balances: A window into streamflow variability over a range of time scales, Water Resour. Res., 50, 5638–5661, doi:10.1002/2014WR015692.

- Berghuijs, W. R., A. Hartmann, and R. A. Woods (2016), Streamflow sensitivity to water storage changes across Europe. Geophys. Res. Lett., 42, doi: 10.1002/2016GL067927.

- Harman, C. J., Troch, P. A., & Sivapalan, M. (2011). Functional model of water balance variability at the catchment scale: 2. Elasticity of fast and slow runoff components

to precipitation change in the continental United States. Water Resources Research, 47(2).

- Chen, X., N. Alimohammadi, and D. Wang (2013), Modeling interannual variability of seasonal evaporation and storage change based on the extended Budyko framework, Water Resour. Res., 49, 6067–6078, doi:10.1002/wrcr.20493.

- Renner, M. and Bernhofer, C.: Applying simple water-energy balance frameworks to predict the climate sensitivity of streamflow over the continental United States, Hydrol. Earth Syst. Sci., 16, 2531-2546, doi:10.5194/hess-16-2531-2012, 2012.

- Sivapalan, M., Yaeger, M. A., Harman, C. J., Xu, X., & Troch, P. A. (2011). Functional model of water balance variability at the catchment scale: 1. Evidence of hydrologic similarity and space‐time symmetry. Water Resources Research, 47(2).

- Vano, J. A., B. Nijssen, and D. P. Lettenmaier (2015), Seasonal hydrologic responses to climate change in the Pacific Northwest, Water Resour. Res., 51, 1959–1976, doi:10.1002/2014WR015909.

- Xu, X., Liu, W., Rafique, R., & Wang, K. (2013). Revisiting continental US hydrologic change in the latter half of the 20th century. Water resources management, 27(12), 4337-4348.

---

## Editor Comment (EC1) · H. Li (Editor) · 17 Feb 2016

Thanks Wouter for the excellent comments. The authors are encouraged to respond and address his comments seriously.
* * *

---

## Referee Comment (RC1) · M.C. Westhoff (Referee) · 9 Mar 2016

General comments

In this manuscript a three-parameter elasticity model is described and applied to 245 MOPEX catchments. Starting point was the elasticity model of Arora (2002) which has been extended by adding an elasticity value for storage change. This extended model was first compared with that of Arora (2002) at annual time scales after which elasticities at seasonal scale were investigated: A time scale which is too short to investigate with the model of Arora (2002).

However, in my opinion this manuscript is not ready for publication yet. The main points

are listed below:

1) It is correct that at seasonal time scales, storage cannot be neglected, but I have some problems with how this is incorporated in Eq. (4). In this equation Q, PET and P are long term average fluxes, which are not very sensitive when long time series are one year longer or shorter. However, storage change (DS) is a state variable and is given by DS = S(t=n) − S(t=0), where n denotes the length of the time series. Because storage oscillates around zero (in a steady climate), DS is relatively large if n = x years + 6 months and small if n = x year + 0 months: In fact each year there will be a moment in time when DS is zero. This shows that DS is very sensitive to the exact length of the time series. And if it is zero, the last term in Eq (4) is divided by zero leading to infinity. A possible way to overcome this sensitivity to n may be to use the standard deviation of DS (although this may give problems on seasonal time scales since the sign of the change disappears in standard deviations).

2) The seasons are defined as 3 month averages, which is indeed a logical thing to do. However, I am missing a sensitivity analysis on the effect of changing these three month averages with a couple of days or weeks. Also, how do the separation of the seasons correspond to the (start of the) hydrological year of each catchment.

3) A thorough discussion is missing: Especially about the meaning of all the seasonal elasticities: Why is it useful to know them, what do they say about the hydrology of a certain catchment, how sensitive are they to measurement errors, what is the influence on snow, etc. Please couple back to the (in my opinion main-) goal of the paper, which is listed on page 3, Line 1: "[it] would serve the purpose of understanding the climate and physical controls".

Specific comments

1) I do agree with the comment posted by Wouter Berghuijs

2) Section 3.4: The description of all results reads as a long list of numbers. I suggest

highlighting the meaning of the individual results and instead of stating that a certain region (e.g. western part of USA) has a certain elasticity, cluster these results in more hydrological terms, such as e.g. the snow dominated catchments have an elasticity of ...

3) P9, L12-13: "this increase ... the same season". This is a strong statement: is there any proof for this?

4) P12, L12-13: "This suggests ... of the basins". This is a strong statement: is there any proof for this?

5) Conclusions: only point a) is a conclusion. Point b,c and d just summarize the 'observations'.

Technical corrections

1) P4, L11: To me it is not an empirical formula, but simply the definition of elasticity

2) For all symbols: use only one letter plus subscripts, since e.g. PET can also be interpreted as P times E times T.

3) P7, L4: What is meant with 'irrespective of the sign'

4) P7, L22: refer to figure 5 instead of 4

5) the paragraph "Streamflow elasticity due to Potential evapotranspiration:"starting at page 11, contains several sloppy typos. Please check carefully.

6) add units to all axes and colour bars of the figures.

References

Arora, V. K.: The use of the aridity index to assess climate change effect on annual runoff, Journal of Hydrology, 265, 164-177, 2002.

---

## Referee Comment (RC2) · Anonymous Referee #2 · 22 Mar 2016

This manuscript presented a three parameter streamflow elasticity model to relate streamflow change with changes in precipitation, potential evapotranspiration, and storage. It is an interesting attempt. However, some improvements are required, and the comments and suggestions are given as below. Therefore, I recommend a major revision.

Major comments:

1. The authors used equation (4) to evaluate the impact on streamflow from precipitation, potential evapotranspiration, and storage change. In this equation, the authors estimated storage change as "(DSt - DS)/DS". I don't think that it is a good choice. According to the definition of this manuscript, DS is the long-term average of storage

change, and it means that DS generally approaches zero in many basins (if there is no storage change). Therefore, it will lead to infinity for the third term on the right side of equation (4). In addition, the sign of storage change elasticity depends on the sign of DS. Consequently, we can't judge whether increasing storage leads to decreasing streamflow according to positive storage change elasticity. In that case, I suggest using storage replacing DS, or using storage change replacing "(DSt - DS)/DS".

2. The structure of this manuscript. In Section 3, the first paragraph represents how to obtain the results. It is better to remove it into Section Methodology. Similarly, first paragraph of Section 3.5 should be removed. In P.4, the sentences from line 9-21 review the researches on the elasticity, and it is better to remove them into the Section Introduction.

3. Figure 8 shows that the potential evapotranspiration elasticity is larger than 0 in some basins and less than 0 in the other basins. It indicates that increasing potential evapotranspiration leads to increasing streamflow in some basins but leads to decreasing streamflow in the other basins. On the causes for the opposite impacts on streamflow, more explanations and discussions are required.

4. Figure 9 shows that the storage change elasticity is larger than 0 for many basins in spring and summer. It means that declining storage will lead to a decreasing streamflow in those basins. At the same time, the storage change elasticity is less than 0 for other basins in spring and summer, which means declining storage resulting in increasing streamflow. The underlying mechanisms of the phenomenon should be explained and discussed.

Minor comments:

1. On the meanings of AIC and BIC, more explanations are required, i.e. why "the preferred model is the one in which the AIC value would be minimum."

2. P.2, line 5-6, Wand and Wang (2011) should be Yang and Yang (2011).

3. P.3, line 8, please check the reference Jiali et al., 2014.

4. Figure 1, the unit of the legend is missing.

5. P.2, line 19, P.7, line6, and so on, "lesser" should be "less".

———————————————————

---

## Author Comment (AC1) · 11 May 2016

**Response to Dr. Berghuijs**

We very much appreciate Dr. Berghuijs for his valuable comments that helped us to improve the manuscript. Note: The text provided in italics will be incorporated in the revised draft.

**Suggestion#1:**

"Later, Dooge (1992) used a hortonian approach to quantify. . .' is a very vague description of what Dooge did. Essentially Dooge derived streamflow sensitivities based on different Budyko type equations (Ol'dekop, Schreiber, Budyko, etc etc). I think therefore this needs to be clarified compared to the current version of the text. He did something very similar to Arora (2002), but Dooge did not yet decomposed the role of climate into P and PET, and considered them as a lumped parameter (P/PET)

**Author's reply:** We do agree with the reviewer that Dooge derived stream flow sensitivities based on different Budyko equations. But, in section (5) of Dooge [1992] article, he mentioned that Hortonian approach is utlised to analyse the sensitivity of runoff to climate change. Considering the suggestion, we have incorporated the following information in to revised manuscript:

*"Later Dooge (1992) devised a method to quantify sensitivity of streamflow to both precipitation and PET. Arora, (2002) extended this work of assessing streamflow sensitivities to PET and P utilizing General circulation model (GCM) data."*

**Suggestion #2:**

"All the elasticity-based models have shown that precipitation has a greater positive influence on streamflow" is not clearly formulated. Do you mean "All the elasticity based models have shown that precipitation elasticity is positive"?

**Author's reply:** we meant that most of the studies point to a conclusion that precipitation has more influence on streamflow when compared to PET, temperature, wind speed etc.  Our modified text would be, *"Most of the annual elasticity-based model studies point to a common conclusion of higher precipitation influence on streamflow when compared to other climate variables like PET, temperature, wind speed etc."*

**Suggestion #3:**

"Fu et al., (2007) suggested that an increase in precipitation along with a positive deviation in temperature would result in lesser impact in streamflow, whereas a negative temperature deviation would result in a higher impact in streamflow" is also unclear. If you formulate these

influences in terms of elasticity values (see previous comment) your statement will get more clear.

**Author's reply:** The above statement will be modified to *"Fu et al., (2007) indicated that in locations with low temperatures, the streamflow elasticity to precipitation is higher than locations with higher temperatures."*

**Suggestion #4**

"Yang and Yang (2011) has identified that relative humidity has a positive influence, whereas net radiation and wind speed have a negative influence on streamflow. More recently, Andréassian et al., (2015) has identified a negative influence of potential evapotranspiration on streamflow." Idem (see point 2&3)

**Author's reply:**   the above statement will be modified to *"Yang and Yang (2011) has identified positive and negative stream flow elasticity due to relative humidity and wind speed respectively. More recently, Andréassian et al., (2015) has identified a negative elasticity due to potential evapotranspiration."*

**Suggestion #5:**

"The hydrometeorological data (1948 to 2003) were collected from the Model parameter estimation experiment (MOPEX) basins located in USA, which are considered unaffected by human influence." What do you mean by "unaffected by human influence"; many of these catchments have land-surface conditions that are strongly affected by humans? I.e. catchment in the Midwest are mostly agricultural. Can you specify what you mean by "unaffected by human influences."

**Author's reply:** We do acknowledge that MOPEX basins are somewhat affected by dams and croplands.  Hence, we will change the statement to limited anthropogenic influence assuming minimal human influence.  The following change would be made in the description of the dataset.

*"The hydrometeorological data (1948 to 2003) were collected from the Model parameter estimation experiment (MOPEX) basins located in USA, which are considered to have limited human influence [Schaake et al.,2006], which allows this study to focus on seasonal climate controls."*

"Therefore, there is an opportunity to investigate the elasticity of streamflow at the seasonal scale to explore the seasonal control of climate on water resource availability." Please not that your concept of seasonal elasctity values is not per se novel. See e.g. Vano et al., 2015.

**Author's reply:** Yes, we do agree with reviewer that the seasonal elasticities concept is not novel. However, our objective is to improve our understanding of seasonal elasticities by utilizing soil water storage as well as the the covariation of precipitation, potential evaporation and storage change in determination of seasonal elasticities. Hence to make it clear, we have modified our introduction section in revised manuscript as follows:

*Most of the elasticity models are applied at annual scales, however, the dominant control of climatic and landscape properties on hydrologic responses are time scale dependent (Atkinson et al., 2002; Farmer et al., 2003; Wang and Alimohammadi, 2011). Estimating this seasonal control of climate on stream flow can be beneficial to water resources managers and planners. The water availability and demand change across each season and as a water resource manager or planner, it is very important to balance these needs by constructing storage facilities or by implementing efficient water conservation practices. However, before implementing these strategies, we first need to understand how different climate factors affect stream flow at seasonal scales in conjunction. In this direction, previous studies (Vanos et al., 2014, Guo et al., 2008, Berguijis et al., 2014, Berguijis et al., 2016; Chen et al., 2013; Istanbulluoglu et al., 2012; Jiang et al., 2015; Ye et al., 2015;) have investigated water balance dynamics by considering seasonality, storage change and extremes. However, these studies have not investigated the combined effect of various climate factors on streamflow at a seasonal scale. A conjunct analysis would likely to provide a more robust solution by considering the coevolution of elasticities and climate variables. As discussed above, climate elasticity provides an easy way to integrate the effects of various climate factors on streamflow without directly considering the effects of soil, land cover etc. For example, a positive precipitation elasticity value of 2 indicates a 2% increase of stream flow with 1% increase in precipitation, whereas a negative storage change elasticity value of 2 indicates a 2% decrease in stream flow with 1 % increase in ground water storage. Further, several studied have explored the relationship between mean annual catchment properties and the elasticities[ Sankarasubramanium et al., 2001; Chiew, 2006; Fu et al., 2011; Sun et al., 2013]. A similar exploration extended to seasonal scale would further assist the planners to create a catchment scale strategy for efficient management of seasonal water resources. Hence, a natural extension of this climate elasticity framework to a seasonal scale would serve our purpose of understanding the seasonal climate and physical controls on water resource availability.*

*Usually, most of the climate elasticity models assume that at annual scale both water storage change and groundwater loss are insignificant (Yang and Yang, 2011; Arora 2002). This assumption leads to a simplified water balance equation, which represents precipitation as a sum of evapotranspiration and streamflow. But, this assumption holds true only if the deep ground water storage is negligible over the considered time period for annual studies (Wang, 2014; Tomer and Schilling, 2009). Therefore, we also check the validity of this assumption by including a term of ground and soil water storage at annual scale. Similarly, at a seasonal scale also these changes cannot be neglected. Hence, the purpose of the article is threefold – (a) Testing the performance of elasticity model at annual scale by incorporating storage change as an influencing component; (b) to evaluate climate elasticities at the seasonal scale, and (c) to explore the relationships between estimated elasticities and catchment properties.*

*The manuscript is organized as follows: in section 2, data and methodology were discussed. Section 3 discusses the results by evaluating the modified climate elasticity model at an annual scale by incorporating precipitation, potential evapotranspiration and change in storage components. Further, we present the stream flow elasticity at a seasonal scale and evaluate their spatial variability. Finally, section 4 presents the conclusions along with the implications of these results.*

**Suggestion #7:**

How does snow influence your study? Snow strongly affects your seasonal and annual water balances (Berghuijs et al., 2014, 2014). I suspect for example that snow links to your statement: "There appears to be a lag in the response of streamflow to rainfall with the high elasticity values starting in winter in the western part of USA. However, it also appears to follow a cycle similar to what we have seen in the eastern part of USA. This clearly highlights the differential behavior of western and eastern USA streamflow elasticities due to precipitation."

Author's reply: The author's would like to thank the reviewer for this suggestion. We checked whether snow has influenced the seasonal elasticities for the basin western USA where snow fraction is greater than 0.15 as outlined by *Berguijis* et al., 2014. Taking cue from this and other related studies, we found out that snow influences stream flow elasticity in the pacific northwestern region. Keeping these things in mind we are incorporating the following changes in the manuscript.

*Overall, as previously put forth by numerous studies in case of annual water balances, comparatively precipitation has higher elasticity values when compared to both PET and Storage changes even at seasonal scale. Considerable seasonality of rainfall elasticity is observed in most of the MOPEX basins in USA. However, the catchments in eastern USA exhibit*

*contrasting features of less rainfall seasonality but more streamflow seasonality (Supplementary figures). This suggests a prominent role of DS, AET and PET in streamflow seasonality since human influence is considered minimal in the eastern region (Wang and Hejazi, 2011). Another important observation is that the lag time exhibited by the catchments in western USA in terms of precipitation elasticity. There appears to be a precipitation plus snowfall excess during fall and streamflow excess during spring. Whereas, during winter, the precipitation plus snowmelt is in phase with streamflow during winter (Berghuilius 2014). This might be the reason for the high elasticities during winter. However, this result should be interpreted with caution, since the western USA has significant human induced streamflow changes (Wang and Hejazi, 2011). Also, the storage changes have shown considerable seasonal elasticity values. The seasonal DS elasticities indicate that soils act as a natural reservoir and subsequently supply and store the streamflow during various seasons. For example, during the higher water demand in summer, the ground water (storage) supplies water to the streamflow resulting in a positive elasticity in most of the MOPEX basins. Whereas during winter and spring, the soil gets recharged and that leads to negative elasticity values. However, we observed that in western USA, the negative elasticity magnitude peaks during winter unlike the rest of US MOPEX basins. This may be mainly because groundwater contribution to streamflow is inversely correlated to snowmelt runoff (Huntington and Niswonger, 2012). Hence, it possibly has high negative elasticity values when the snow accumulates in winter. Whereas, when the snowmelt runoff starts in the spring it starts contributing to streamflow indicating positive elasticities.*

**Suggestion #8:**

One of the disadvantages of your approach is that the elasticity to water storage changes is derived from all residual (and uncertain) other data sources (Q, P, AET). Especially AET is uncertain as this cannot be directly measured. In the meanwhile there is also a way to calculate this metric using hydrograph recession analyses, see Berghuijs et al. (2016). Are you happy with your current approach or do you think that this method could actually make your results more robust?

**Author's reply:** *Berguijis   et al. (2016)* has derived storage sensitivity of streamflow using hydrograph recession methods built upon an analytical approximation which assumes that water storage is the only source of streamflow [Brutsaert and Nieber,1977]. The hydrographs are selected for winter season only to reduce the influence of evapotranspiration.   In this article we considered that runoff is sensitive to a combination of climate factors and accounted for the covariation of precipitation, potential evaporation and storage change. Also, as we are considering all seasons, we cannot neglect the evapotranspiration component. . Hence, we recognize that both issues are different and hence both have to be seen as different contributions.

Obviously, even our study has its own limitations due to the use of satellite evapotranspiration dataset which is likely to have its own uncertainies.

**Sugestion #9:**

I think the following statement is very specultive "We can see that during fall, the eastern region has a negative elasticity indicating a decrease in stream flow due to increase in potential evapotranspiration. But, in the south western watersheds we can see a positive elasticity value indicating an increase in stream flow due to potential evapotranspiration. This increase can be viewed as an increase in available moisture locally causing more rainfall and subsequently more rainfall within the same season. This contrasting behaviour might be due to higher temperatures in southern USA increasing the potential evapotranspiration and thus the capacity to withhold moisture. This might be similar to the precipitation recycling concept introduced by Eltahir and Bras [1998]." Do you have more evidence to support this?

**Author's reply:**

This is a very relevant comment which requires a dedicated and separate study. However to support our findings we are including the following discussion in the revised manuscript

"*In previous studies also, certain catchments have shown positive streamflow elasticities due to potential evapotranspiration [Andréassian et al., 2015, Yang et al., 2014]. The positive PET elasticity may be caused by the local climate feedback. According to previous studies (e.g., (Koster et al. 2004; Guo et al., 2006 Mei and Wang, 2011), the central USA has strong land-atmosphere coupling strength. The PET plays an important role in the linkage of soil moisture and precipitation in the land-atmosphere interactions. Based on the positive land-atmosphere interactions, the increased soil moisture would lead to a cascading effect of increase of temperature (indirectly PET) and precipitation. The increased precipitation would therefore lead to the increase of Streamflow. In this notation, the PET has a positive relationship with precipitation, which would lead to a positive PET elasticity. The positive PET elasticity are within these hotspots in summer season*".

**Suggestion #10:**

Can you clarify your interpretation for: "Figure 8 illustrates the seasonal pattern of streamflow elasticities due to storage change. It was observed that the seasonal elasticities exhibit change in spatial clusters. For example, the eastern USA seems to exhibit a cycle of negative elasticities in fall, and then its intensity decreased in winter, becomes almost negligible in spring and exhibits positive elasticity in summer. However, the watersheds in south eastern coast seem to exhibit negative elasticities in summer followed by a decrease in negative elasticity values in fall and

winter. This region exhibits positive elasticity values in spring whereas the rest of eastern USA exhibits positive elasticity in later season. S" Do you expect that this is the actual physical behavior of the catchment or can these seasonal changes also be induced by the potential bias introduced in your result due to uncertainties in the components of water balances?

**Author's reply**: This suggestion is certainly interesting. We selected one MOPEX basin in the southern region of Florida and two basins in the state of New Mexico with the following basin ids, 94975000, 2273000 and 2296750 respectively. We investigated the summer season flows, since we suspected some anomalous behavior due to their negative elasticity values. We plot the seasonal averages of the selected time period. The streamflow and evapotranspiration are lower than rainfall amounts. The values seem normal and do not indicate an anomalous behavior. However, we do acknowledge the fact that the streamflow in those catchments is influenced by storage facilities (Wang and Hejazhi, 2012), therefore additional research is expected to address whether this is a natural behavior of the catchment.

[Figure]

**Suggestion #11:**

I do not think that the analysis of catchment properties influence on elasticity's is done rigorously. Can you make this part of the analysis a bit more appealing and convincing? Also, why did you choose these catchment properties?

**Author's reply:** Based on the reviewers suggestion we have quantified the relationship using linear and nonlinear association metrics between seasonal elasticities and catchment properties. We have included this analysis as a part of our objective too. We have made the following changes at appropriate sections in the revised paper.

 " *Studies [Sankarasubramanium et al., 2001; Chiew, 2006; Fu et al., 2011; Sun et al., 2013]  estimated that the there exists a nonlinear relation between the annual elasticities and the considered hydroclimatic variables. Expecting a similar behaviour at seasonal scale, we quantify the strength of association, using*

*both linear and nonlinear association metrics. For the purpose of estimating the linear and nonlinear associations we considered the seasonal precipitation (P), Storage Changes (DS), Potential evapotranspiration (PET), Aridity Index (AI) and evaporative index (EI). Even though we have estimated the elasticities based on seasonal variations in P, Q and DS, we want to further explore the relationship between seasonal magnitudes of these variables and the calculated elasticites. In addition to that, aridity index (AI) and evaporative index (EI) which are indicators of catchment (climate) and physical characteristics can explore the seasonal control of catchment properties on elasticities. Hence, we aggregate P, Q, DS, PET and AET at seasonal scales and calculate their averages over the study period. From those averages, seasonal AI and EI are estimated as PET/ (P-DS) and AET/ (P-DS) respectively (Chen et al, 2012). We estimated the linear association based on Pearson correlation coefficients and estimated the level of significance based on p values derived from two sided permutation test of 999 replicates (Helsel and Hirsch, 1992). Several nonlinear association metrics like mutual information (MI)(Cover and Thomas , 1991), Maximal information coefficient (MIC)(Reshef DN, et al. 2011), Hoeffding distance [Hoeffding, 1948] and distance correlation [Szekely and Rizzo, 2009] are prevalent in literature. Among these measures, distance correlation coefficient is easier to implement and has comparatively better statistical power [Kinney and Atwal, 2014], which is used in this study. As this metric is new to field of hydrology, we present the derivation in the following text.*

*For computing the distance correlation measure between two random variables (X, Y), we first compute the pairwise distances matrices ($a_{i,j}$) and ($b_{i,j}$) as*

$$a_{i,j} = \left\| X_i - X_j \right\|$$

*(1)*

$$b_{i,j} = \left\| Y_i - Y_j \right\|$$

*(2)*

*where i, j = 1,2,3,4,5,....n and $\left\| \cdot \right\|$ denotes the Euclidean (in our case) distance. Now, we center these distances matrices as shown below*

$$A_{i,j} = a_{i,j} - \overline{a}_{i.} - \overline{a}_{.j} + \overline{a}_{..}$$

*(3)*

$$B_{i,j} = b_{i,j} - \overline{b}_{i.} - \overline{b}_{.j} + \overline{b}_{..}$$

*(4)*

*where, $\overline{a}_{i.}, \overline{b}_{i.}$ are the $i^{th}$ row means, $\overline{a}_{.j}, \overline{b}_{.j}$ are the $j^{th}$ column means, $\overline{a}_{..}, \overline{b}_{..}$ are the overall mean of the ($a_{i,j}$) and ($b_{i,j}$) matrices, respectively. Then, we estimate the square of distance covariance as the arithmetic average of the products $A_{i,j}$ and $B_{i,j}$.*

$$dCov_n^2(X,Y) = \frac{1}{n^2} \sum_{i,j=1}^{n} A_{i,j}.B_{i,j}$$

(5)

Similarly, we estimate the distance variance as

$$dVar_n^2(X) = dCov_n^2(X,X) = \frac{1}{n^2} \sum_{i,j=1}^{n} A_{i,j}^2$$

(6)

Finally, the distance correlation is obtained as

$$dCor(X,Y) = \frac{dCov(X,Y)}{\sqrt{dVar(X) \times dVar(Y)}}$$

(7)

*The significance of the calculated correlation is estimated by one sided permutation test of 999 replicates. In both the linear and nonlinear cases, only relations which satisfy the 95% significance level ($p < 0.05$) are presented. "*

We have removed the previous discussion and provided a revised discussion based on our new findings, as discussed below.

*"This analysis allows for a quantitative investigation of relations between the seasonal elasticities and catchment climate properties and gives an understanding of the possible governing factors. Figure (shown below), shows the statistically significant linear (top panel) and nonlinear correlations (bottom panel) between the considered the seasonal hydroclimatic variables and elasticities. We excluded high elasticity values greater than 10 and lesser than -5 in this analysis which may be unrealistic due to uncertainity in the data sources by visual examination of the scatterplots provided in the supplementary information. As we expected, there does exist significant nonlinear associations between elasticities and considered catchement properties. Hence, we base most of our discussions in this text on the nonlinear associations presented in the figure, but sometimes refer the linear association for determining the directionality of the relationships.*

*It is interesting to see how the hydroclimatic variables relationship changes with each season. For example, during summer there exists a stronger association of rainfall magnitude and less predominant association of streamflow with elasticities than in other seasons. During summer, due to relatively high temperatures and inadequacy of available water as streamflow, the catchments become water limited leading to be more dependent on rainfall as a source of water. This behaviour is more prevalent in storage changes elasticity. Also, it is obvious that the elasticities are more governed by the magnitudes of streamflow in most of the other cases. But, the linear associations suggest that the streamflow is inversely proportional to precipitation and potential evapotranspiration elasticities. Usually, if the catchments with high streamflow are highly elastic in nature, even minimal amount of rainfall would result in high*

*streamflow generation, which might impact existing flood and water management activities. Hence, this inverse relationship which is achieved either through artificial/natural storage facilities is beneficial to water management. In the case of elasticity due to storage change, when the elasticities have negative values(in fall and winter), there exist a positive linear relationship with streamflow achieving a similar goal of efficient water management. However, we suspect that this might not be a natural behavior of a catchment as significant human interference might have created this behaviour (Wang and Hezaidi, 2012; Ye et al.,2014). Also, there exists a significant inter-relationship between the hydroclimatic variables and determined elasticities. For example, the seasonal magnitude of DS affects PET elasticity as well as precipitation elasticity in most of the seasons. Same conclusion can be arrived in other cases too.*

*The aridity index (AI), which is a possible indicator of catchment &climate (higher the aridity index, drier is the catchment) [Jones et al., 2012] also has significant association with climate elasticities. The negative correlations between AI and DS elasticities indicate that the dry catchment have higher DS elasticities. Hence, drier catchments have the capacity to store streamflow during wet seasons and aid in streamflow generation during dry seasons. This study could further help in investigating the discharge and recharge mechanisms of the available MOPEX basins. Similarly, interpretations can be made in terms of precipitation elasticity for positive correlations. In addition to that, AI plays a more significant role in spring season, indicating that the elasticities are more susceptible to catchment (climate) conditions in that season. Similarly, the evaporative index which is an indirect gauge of the physical properties of catchments [Jones et al., 2012] has significant associations as well as higher magnitude in the spring season. This analysis complements many studies which have linked the catchment properties at different scales to streamflow dynamics (Chiverton et al., 2015; Ann vann loon et al., 2015; Gaal et al., 2012; Ye et al., 2015). However, we do not want to stress on a single dominant factor affecting the streamflow elasticities, since there appears to be a strong interplay between elasticites and all the considered catchment properties with substantial seasonal variations."*

[Figure]

***Figure***: *The linear and nonlinear association strengths as determined by Pearson and distance correlation coefficients.[Note: In the figure, we have sorted the strength of association separately for each season and the hydroclimatic variables are represented by different colors and only statistically significant [p<0.05] correlation strengths are shown here.]*

**Suggestion #12**

The discussion part of this paper is somewhat thin in my opinion.

Authors, reply: we have modified the manuscript to improve the discussion. **Reviewer's Suggestion #13**

Figure 1 & 5: can you please be accurate in what we are looking at, and specify the units (even if they're dimensionless).

Author's reply: We have indicated the dimensions in figure 1. But, we have included that the elasticity differences are dimensionless in the text.

[Figure]

Fig 1: Mean of annual Precipitation, Potential evapotranspiration, Streamflow and Storage changes in mm/day from 1983 to 2003.

[Figure]

Figure 5: The difference between trivariate and bivariate precipitation elasticities (dimensionless). On the right side, a violin plot showing the distribution of these differences.

**Suggestion #14.**

Figure 6: On the left side? Or on right side?

Author's reply: [It is on the right side.]The change will be made in the revised manuscript.

**Suggestion #15.**

Figure 8: this colorbar makes it impossible to read the figure well

Author's reply: We have removed the state boundaries and inverted the colors to incorporate the changes.

[Figure]

Figure 8: Seasonal distribution of streamflow Elasticity due to PET (dimensionless)

**References:**

Andréassian, V., Coron, L., Lerat, J. and Le Moine, N.: Climate elasticity of streamflow revisited–an elasticity index based on long-term hydrometeorological records, Hydrology and Earth System Sciences Discussions, 12, 3645-3679, 2015.

Arora, V. K.: The use of the aridity index to assess climate change effect on annual runoff, Journal of Hydrology, 265, 164-177, 2002.

Atkinson, S., Woods, R. and Sivapalan, M.: Climate and landscape controls on water balance model complexity over changing timescales, Water Resour. Res., 38, 50-1-50-17, 2002.

Berghuijs, W. R., Sivapalan, M., Woods, R. A., & Savenije, H. H. (2014). Patterns of similarity of seasonal water balances: A window into streamflow variability over a range of time scales. *Water Resources Research*, *50*(7), 5638-5661.

Berghuijs, W. R., Woods, R. A., & Hrachowitz, M. (2014). A precipitation shift from snow towards rain leads to a decrease in streamflow. Nat. Clim. Change, 4(7), 583-586.

Berghuijs, W. R., A. Hartmann, and R. A. Woods (2016), Streamflow sensitivity to water storage changes across Europe. Geophys. Res. Lett., 42, doi: 10.1002/2016GL067927.

Chen, X., Alimohammadi, N. and Wang, D.: Modeling interannual variability of seasonal evaporation and storage change based on the extended Budyko framework, Water Resour. Res., 49, 6067-6078, 2013.

Chiew, F. H.: Estimation of rainfall elasticity of streamflow in Australia, Hydrological Sciences Journal, 51, 613-625, 2006.

Chiverton, A., Hannaford, J., Holman, I., Corstanje, R., Prudhomme, C., Bloomfield, J., & Hess, T. M. (2015). Which catchment characteristics control the temporal dependence structure of daily river flows?. *Hydrological Processes*, *29*(6), 1353-1369.

Cover, T. M., & Thomas, J. A. (1991). Information theory and statistics.*Elements of Information Theory*, 279-335.

Dooge, J. C.: Sensitivity of runoff to climate change: A Hortonian approach, Bull. Am. Meteorol. Soc., 5 73, 2013-2024, 1992.

Farmer, D., Sivapalan, M. and Jothityangkoon, C.: Climate, soil, and vegetation controls upon the variability of water balance in temperate and semiarid landscapes: Downward approach to water balance analysis, Water Resour. Res., 39, 2003.

Fu, G., Charles, S. P. and Chiew, F. H.: A two-parameter climate elasticity of streamflow index to assess climate change effects on annual streamflow, Water Resour. Res., 43, 2007.

Gaál, L., Szolgay, J., Kohnová, S., Parajka, J., Merz, R., Viglione, A., & Blöschl, G. (2012). Flood timescales: Understanding the interplay of climate and catchment processes through comparative hydrology. *Water Resources Research*, *48*(4).

Guo, J., Li, H., Leung, L. R., Guo, S., Liu, P. and Sivapalan, M.: Links between flood frequency and annual water balance behaviors: A basis for similarity and regionalization, Water Resour. Res., 50, 937-20 953, 2014.

Helsel, D. R., & Hirsch, R. M. (1992). *Statistical methods in water resources*(Vol. 49). Elsevier.

Hoeffding, W. (1948). A non-parametric test of independence. *The Annals of Mathematical Statistics*, 546-557.

Huntington, J. L., & Niswonger, R. G. (2012). Role of surface-water and groundwater interactions on projected summertime streamflow in snow dominated regions: An integrated modeling approach. *Water Resources Research*, *48*(11).

Istanbulluoglu, E., Wang, T., Wright, O. M. and Lenters, J. D.: Interpretation of hydrologic trends from a water balance perspective: The role of groundwater storage in the Budyko hypothesis, Water Resour. Res., 48, 2012.

Jiang, C., Xiong, L., Wang, D., Liu, P., Guo, S. and Xu, C.: Separating the impacts of climate change and human activities on runoff using the Budyko-type equations with time-varying parameters, Journal of Hydrology, 522, 326-338, 2015.

Jones, J. A., Creed, I. F., Hatcher, K. L., Warren, R. J., Adams, M. B., Benson, M. H., ... & Clow, D. W. (2012). Ecosystem processes and human influences regulate streamflow response to climate change at long-term ecological research sites. *BioScience*, *62*(4), 390-404.

Kinney, J. B., & Atwal, G. S. (2014). Equitability, mutual information, and the maximal information coefficient. *Proceedings of the National Academy of Sciences*, *111*(9), 3354-3359.

Reshef, D. N., Reshef, Y. A., Finucane, H. K., Grossman, S. R., McVean, G., Turnbaugh, P. J., ... & Sabeti, P. C. (2011). Detecting novel associations in large data sets. *science*, *334*(6062), 1518-1524.

Sankarasubramanian, A., Vogel, R. M. and Limbrunner, J. F.: Climate elasticity of streamflow in the United States, Water Resour. Res., 37, 1771-1781, 2001.

Schaake, J., S. Cong, and Q. Duan (2006), The US MOPEX data set, IAHS Publ., 307, 9–28.

Székely, G. J., & Rizzo, M. L. (2009). Brownian distance covariance. *The annals of applied statistics*, *3*(4), 1236-1265.

Tomer, M. D., & Schilling, K. E. (2009). A simple approach to distinguish land-use and climate-change effects on watershed hydrology. *Journal of Hydrology*, *376*(1), 24-33.

Van Loon, A. F., & Laaha, G. (2015). Hydrological drought severity explained by climate and catchment characteristics. *Journal of Hydrology*, *526*, 3-14.

Vano, J. A., Nijssen, B., & Lettenmaier, D. P. (2015). Seasonal hydrologic responses to climate change in the Pacific Northwest. *Water Resources Research*, *51*(4), 1959-1976.

Wang, D. and Alimohammadi, N.: Responses of annual runoff, evaporation, and storage change to climate variability at the watershed scale, Water Resour. Res., 48, 2012.

Wang, D., & Hejazi, M. (2011). Quantifying the relative contribution of the climate and direct human impacts on mean annual streamflow in the contiguous United States. *Water Resources Research*, *47*(10).

Wang, X.: Advances in separating effects of climate variability and human activity on stream discharge: An overview, Adv. Water Resour., 71, 209-218, 2014.

Yang, H. and Yang, D.: Derivation of climate elasticity of runoff to assess the effects of climate change on annual runoff, Water Resour. Res., 47, 2011.

Ye, S., Li, H., Li, S., Leung, L. R., Demissie, Y., Ran, Q. and Blöschl, G.: Vegetation regulation on streamflow intra-annual variability through adaption to climate variations, Geophys. Res. Lett., 2015.

---

## Author Comment (AC2) · 11 May 2016

**Response to Dr. Westoff**

We very much appreciate Dr. Westoff for his valuable comments that helped us to improve the manuscript. Note: The text provided in italics will be incorporated in the revised draft.

**Suggestion #1:**

It is correct that at seasonal time scales, storage cannot be neglected, but I have some problems with how this is incorporated in Eq. (4). In this equation Q, PET and P are long term average fluxes, which are not very sensitive when long time series are one year longer or shorter. However, storage change (DS) is a state variable and is given by DS = S(t=n) − S(t=0), where n denotes the length of the time series. Because storage oscillates around zero (in a steady climate), DS is relatively large if n = x years + 6 months and small if n = x year + 0 months: In fact each year there will be a moment in time when DS is zero. This shows that DS is very sensitive to the exact length of the time series. And if it is zero, the last term in Eq (4) is divided by zero leading to infinity. A possible way to overcome this sensitivity to n may be to use the standard deviation of DS (although this may give problems on seasonal time scales since the sign of the change disappears in standard deviations).

**Author reply:** Yes, we do agree that at annual scale (equation 4), the DS may tent to zero. In this article, we have hypothesized that, if DS is zero, then we can always go back to equation 2 (two parameter equaltion), which is defined for situations were DS is zero. But, the assumption of DS =0 is not always valid. For example, in regions were the anisotropy ratio (Vertical hydraulic conductivity/Horizontal hydraulic conductivity) is not negligible, the ground water losses do occur, indicating that DS ≠0 (Wang, 2014). However, we do agree that the DS is calculated as a residual (P-Q-AET) and likely to have uncertainties due to usage of data from different sources. So, this may result in either underestimation or overestimation of storage change than it is derived in this article. Hence, until more information is available, this can be deemed as a hypothesis that remains to be tested. However, in this study we do neglect the regions for which DS =0 at all scales. But, when it comes to seasonal scales, the DS would theoretically not be zero since, in a particular season; water balance would contain deficits or excesses depending on occurrence of rainfall events and change in temperature.

**Suggestion #2:**

The seasons are defined as 3 month averages, which is indeed a logical thing to do. However, I am missing a sensitivity analysis on the effect of changing these three month averages with a couple of days or weeks. Also, how do the separation of the seasons correspond to the (start of the) hydrological year of each catchment.

Author's reply: This is an interesting suggestion. To investigate the effect of length on seasonal elasticities, we decreased (increased) the length of each season by decreasing (increasing) the number of days from each season and calculated the seasonal elasticities. Then we calculated the coefficient of determination ($R^2$) between elasticities computed with original season lengths and the increased (decreased) season lengths. We limited our modified season lengths to original season length ±7 days. As anticipated, the $R^2$ decrease with increase (decrease) in season lengths. However, the least $R^2$ value obtained was 0.93 in the case of PET elasticity in spring. The average $R^2$ for the elasticities is around 0.99 indicating that small changes in season length may not have significant impact for elasticity estimation.

[Figure]

The United States Geological Survey defines Hydrologic year as the time period between October 1st of one year and September 30th of the next year. However, in our article we have considered seasons as a three month average usually considered in a majority of seasonal studies.

**Suggestion #3:**

A thorough discussion is missing: Especially about the meaning of all the seasonal elasticities: Why is it useful to know them, what do they say about the hydrology of a certain catchment, how sensitive are they to measurement errors, what is the influence on snow, etc. Please couple back

to the (in my opinion main-) goal of the paper, which is listed on page 3, Line 1: "[it] would serve the purpose of understanding the climate and physical controls".

Author's reply: Thank you for the suggestion. We have incorporated required suggestion in the revised draft. In addition to those, we have added the following text to increase the discussion part to couple back to goal of the paper in explaining the climate and physical controls of streamflow.

*"Overall, the low values of stream flow elasticities due to PET have highlighted the fact that PET play less role (Also, less number of statistically significant streamflow elasticities due to PET) in influencing the annual streamflow (Zhao et al., 2009, 2010; Wang et al., 2011). However, PET which is an indirect measure of temperature does indicate lower PET would result in higher precipitation elasticity (Fu et al., 2007). In addition to that, we observed that the modified elasticity model clearly strengthens the inter-relationship between precipitation, potential evapotranspiration, stream flow and storage changes. This would eventually point to a prominent role of storage changes in the generation of streamflow at annual scales as concluded in other studies (Wang et al., 2009; Gonzalo and Fan et al., 2012; Huntington and Niswonger, 2012). Hence, neglecting these changes would result in either underestimation or overestimation of precipitation and PET elasticities. Moreover, in the situation where DS = 0, we can always go back to equation 2 which neglects the effect of DS on annual streamflow. However, even though the trivariate elasticity model performs better than the bivariate model, we can see that DS is calculated as a residual (P-Q-AET) and likely to have uncertainties due to usage of data from different sources. So, this may result in improper assessment of storage change. Hence, until high quality information with minimum uncertainty in the data sources is obtained, this has to be viewed as a hypothesis that remains to be tested."*

*"Overall, as previously put forth by numerous studies in case of annual water balances, precipitation has higher magnitude elasticity values than compared to both PET and Storage changes even at seasonal scale. Considerable seasonality of rainfall elasticity is observed in most of the MOPEX basins in USA. However, the catchments in eastern USA exhibit contrasting features of less rainfall seasonality more seasonal behavior in streamflow (Supplementary figures). This suggests a prominent role of DS and PET in streamflow seasonality since human influence is considered minimal in the eastern region (Wang and Hejazi, 2011). Another observation worth mentioning, is the lag exhibited by the catchments in western USA in terms of precipitation elasticity. There appears to be a precipitation plus snowfall excess during fall and streamflow excess during spring. Whereas, during winter, the precipitation plus snowmelt is in phase with streamflow (Berghuilius 2014). This might be the reason for the higher elasticities during winter. However, this result should be interpreted with caution, since the western USA*

*has significant human induced changes on streamflow characteristics (Wang and Hejazi, 2011). Also, the storage change have shown considerable seasonal elasticity values. The seasonal DS elasticities indicate that ground water storage act as a natural reservoir and subsequently supply and store the streamflow during various seasons. For example, during summer when the temperatures are high and water requirement is more, ground water supplies water to the streamflow resulting in a positive elasticity in most of the MOPEX basins. Whereas, in winter and spring the soil gets recharged leading to negative elasticity values. However, we observed that in western USA, the negative elasticity magnitude increases during winter unlike the rest of US MOPEX basins. This may be mainly because groundwater contribution to streamflow is inversely correlated to snowmelt runoff (Huntington and Niswonger, 2012). Hence, it possibly has high negative elasticity values when the snow accumulates in winter. Whereas, when the snowmelt runoff starts in the spring it starts contributing to streamflow indicating positive elasticities."*

*"It is interesting to see how the hydroclimatic variables relationship changes with each season. For example, during summer there exists a stronger association of rainfall magnitude and less predominant association of streamflow with elasticities than in other seasons. During summer, due to relatively high temperatures and inadequacy of available water as streamflow, the catchments become water limited leading to be more dependent on rainfall as a source of water. This behavior is more prevalent in storage changes elasticity. Also, it is obvious that the elasticities are more governed by the magnitudes of streamflow in most of the other cases. But, the linear associations suggest that the streamflow is inversely proportional to precipitation and potential evapotranspiration elasticities. Usually, if the catchments with high streamflow are highly elastic in nature, even minimal amount of rainfall would result in high streamflow hampering efficient disaster and water management activities. Hence, this inverse relationship which is achieved either through artificial/natural storage facilities is beneficial to water management. In the case of elasticity due to storage changes, during the seasons of fall and winter when the elasticities have negative values, there exist a positive linear relationship with streamflow achieving a similar goal of efficient water management practice. However, we suspect that this might not be a natural behavior of a catchment, significant human interference might have created this behavior(Wang and Hejazi, 2012, Ye et al., 2015). Also, there exists a significant inter-relationship between the hydroclimatic variables and determined elasticities. For example, the seasonal magnitudes of DS effects PET elasticity as well as precipitation elasticity in most of the seasons. Same conclusion can be arrived in other cases too.*

*The aridity index, which is a possible indicator of catchment climate (higher the aridity index, drier is the catchment) (Jones et al., 2012) also has significant association with climate*

*elasticites. The negative correlations in the case DS elasticities, indicate that the dry catchment have higher DS elasticities. Hence, drier catchments have the capacity to store streamflow during wet seasons and aid in streamflow generation during dry seasons. An in-depth analysis of this could further help in investigating the discharge and recharge mechanisms of the available MOPEX basins. Similarly, interpretations can be made in terms of precipitation elasticity for positive correlations. In addition to that, AI plays a more significant role in spring season, indicating that the elasticities are more susceptible to catchment climate conditions in that season. Similarly, the evaporative index which is an indirect gauge of the physical properties of catchments [Jones et al., 2012] has significant associations peaking in the spring season. For example, this relationship articulates that an increase in evaporative index is accompanied by an increase in precipitation elasticity indicating that the catchments with more physical control on streamflow generate more streamflow even for small events of rainfall. This analysis complements many studies which have linked the catchment properties at different scales to streamflow dynamics (Chiverton et al., 2015; Ann vann loon et al., 2015; Gaal et al., 2012; Ye et al., 2015). However, we do not want to stress on a single dominant factor affecting the streamflow elasticities, since there appears to be a strong interplay between elasticites and all the considered catchment properties with substantial seasonal variations."*

**Specific comments #1:** I do agree with the comment posted by Wouter Berghuijs:

Author's reply: We have addressed these issues; Please have a look at them.

**Specific comments # 2:**

Section 3.4: The description of all results reads as a long list of numbers. I suggest highlighting the meaning of the individual results and instead of stating that a certain region (e.g. western part of USA) has a certain elasticity, cluster these results in more hydrological terms, such as e.g. the snow dominated catchments have an elasticity of .

**Author's reply:**

We have made the suggested changes throughout the revised manuscript.

**Specific comments # 3:**

P9, L12-13: "this increase ... the same season". This is a strong statement: is there any proof for this?

**Author's reply:**

This is a very relevant comment which requires a dedicated and separate study. However to support our findings we are including the following discussion in the revised manuscript

"*In previous studies also, certain catchments have shown positive streamflow elasticities due to potential evapotranspiration [Andréassian et al., 2015, Yang et al., 2014]. The positive PET elasticity may be caused by the local climate feedback. According to previous studies (e.g., (Koster et al. 2004; Guo et al., 2006 Mei and Wang, 2011), the central USA has strong land-atmosphere coupling strength. The PET plays an important role in the linkage of soil moisture and precipitation in the land-atmosphere interactions. Based on the positive land-atmosphere interactions, the increased soil moisture would lead to a cascading effect of increase of temperature (indirectly PET) and precipitation. The increased precipitation would therefore lead to the increase of Streamflow. In this notation, the PET has a positive relationship with precipitation, which would lead to a positive PET elasticity. The positive PET elasticity are within these hotspots in summer season*".

**Specific comments # 4:**

P12, L12-13: "This suggests ... of the basins". This is a strong statement: is there any proof for this?

Author's reply:

We have revised the manuscript, which was also suggested by Dr. Berghuijs. We have made the following changes at appropriate locations in the revised manuscript.

" *Studies [Sankarasubramanium et al., 2001; Chiew, 2006; Fu et al., 2011; Sun et al., 2013] estimated that the there exists a nonlinear relation between the annual elasticities and the considered hydroclimatic variables. Expecting a similar behaviour at seasonal scale, we quantify the strength of association, using both linear and nonlinear association metrics. For the purpose of estimating the linear and nonlinear associations we considered the seasonal precipitation (P), Storage Changes (DS), Potential evapotranspiration (PET), Aridity Index (AI) and evaporative index (EI). Even though we have estimated the elasticities based on seasonal variations in P, Q and DS, we want to further explore the relationship between seasonal magnitudes of these variables and the calculated elasticites. In addition to that, aridity*

*index (AI) and evaporative index (EI) which are indicators of catchment (climate) and physical characteristics can explore the seasonal control of catchment properties on elasticities. Hence, we aggregate P, Q, DS, PET and AET at seasonal scales and calculate their averages over the study period. From those averages, seasonal AI and EI are estimated as PET/ (P-DS) and AET/ (P-DS) respectively (Chen et al, 2012). We estimated the linear association based on Pearson correlation coefficients and estimated the level of significance based on p values derived from two sided permutation test of 999 replicates (Helsel and Hirsch, 1992). Several nonlinear association metrics like mutual information (MI)(Cover and Thomas , 1991), Maximal information coefficient (MIC)(Reshef DN, et al. 2011), Hoeffding distance [Hoeffding, 1948] and distance correlation [Szekely and Rizzo, 2009] are prevalent in literature. Among these measures, distance correlation coefficient is easier to implement and has comparatively better statistical power [Kinney and Atwal, 2014], which is used in this study. As this metric is new to field of hydrology, we present the derivation in the following text.*

*For computing the distance correlation measure between two random variables (X, Y), we first compute the pairwise distances matrices ($a_{i,j}$) and ($b_{i,j}$) as*

$$a_{i,j} = \left\| X_i - X_j \right\|$$

*(1)*

$$b_{i,j} = \left\| Y_i - Y_j \right\|$$

*(2)*

*where i, j = 1,2,3,4,5,....n and $\left\| \cdot \right\|$ denotes the Euclidean (in our case) distance. Now, we center these distances matrices as shown below*

$$A_{i,j} = a_{i,j} - \bar{a}_{i.} - \bar{a}_{.j} + \bar{a}_{..}$$

*(3)*

$$B_{i,j} = b_{i,j} - \bar{b}_{i.} - \bar{b}_{.j} + \bar{b}_{..}$$

*(4)*

*where, $\bar{a}_{i.}, \bar{b}_{i.}$ are the $i^{th}$ row means, $\bar{a}_{.j}, \bar{b}_{.j}$ are the $j^{th}$ column means, $\bar{a}_{..}, \bar{b}_{..}$ are the overall mean of the ($a_{i,j}$) and ($b_{i,j}$) matrices, respectively. Then, we estimate the square of distance covariance as the arithmetic average of the products $A_{i,j}$ and $B_{i,j}$.*

$$dCov_n^2(X,Y) = \frac{1}{n^2} \sum_{i,j=1}^{n} A_{i,j}.B_{i,j}$$

*(5)*

*Similarly, we estimate the distance variance as*

$$dVar_n^2(X) = dCov_n^2(X,X) = \frac{1}{n^2} \sum_{i,j=1}^{n} A_{i,j}^2$$

*(6)*

*Finally, the distance correlation is obtained as*

$$dCor(X,Y) = \frac{dCov(X,Y)}{\sqrt{dVar(X) \times dVar(Y)}}$$

*(7)*

*The significance of the calculated correlation is estimated by one sided permutation test of 999 replicates. In both the linear and nonlinear cases, only relations which satisfy the 95% significance level (p<0.05) are presented. "*

We have removed the previous discussion and provided a revised discussion based on our new findings, as discussed below.

*"This analysis allows for a quantitative investigation of relations between the seasonal elasticities and catchment climate properties and gives an understanding of the possible governing factors. Figure (shown below), shows the statistically significant linear (top panel) and nonlinear correlations (bottom panel) between the considered the seasonal hydroclimatic variables and elasticities. We excluded high elasticity values greater than 10 and lesser than -5 in this analysis which may be unrealistic due to uncertainity in the data sources by visual examination of the scatterplots provided in the supplementary information. As we expected, there does exist significant nonlinear associations between elasticities and considered catchement properties. Hence, we base most of our discussions in this text on the nonlinear associations presented in the figure, but sometimes refer the linear association for determining the directionality of the relationships.*

*It is interesting to see how the hydroclimatic variables relationship changes with each season. For example, during summer there exists a stronger association of rainfall magnitude and less predominant association of streamflow with elasticities than in other seasons. During summer, due to relatively high temperatures and inadequacy of available water as streamflow, the catchments become water limited leading to be more dependent on rainfall as a source of water. This behaviour is more prevalent in storage changes elasticity. Also, it is obvious that the elasticities are more governed by the magnitudes of streamflow in most of the other cases. But, the linear associations suggest that the streamflow is inversely proportional to precipitation and potential evapotranspiration elasticities. Usually, if the catchments with high streamflow are highly elastic in nature, even minimal amount of rainfall would result in high streamflow generation, which might impact existing flood and water management activities. Hence, this inverse relationship which is achieved either through artificial/natural storage facilities is beneficial to water management. In the case of elasticity due to storage change, when the elasticities have negative values(in fall and winter), there exist a positive linear relationship with streamflow achieving a similar goal of efficient water management. However, we suspect that this might not be a natural behavior of a catchment as significant human interference might have created this behaviour (Wang and Hezaidi, 2012; Ye et al.,2014). Also, there exists a significant inter-relationship between the hydroclimatic variables and determined elasticities. For example, the seasonal magnitude of DS affects PET elasticity*

*as well as precipitation elasticity in most of the seasons. Same conclusion can be arrived in other cases too.*

*The aridity index (AI), which is a possible indicator of catchment &climate (higher the aridity index, drier is the catchment) [Jones et al., 2012] also has significant association with climate elasticities. The negative correlations between AI and DS elasticities indicate that the dry catchment have higher DS elasticities. Hence, drier catchments have the capacity to store streamflow during wet seasons and aid in streamflow generation during dry seasons. This study could further help in investigating the discharge and recharge mechanisms of the available MOPEX basins. Similarly, interpretations can be made in terms of precipitation elasticity for positive correlations.   In addition to that, AI plays a more significant role in spring season, indicating that the elasticities are more susceptible to catchment (climate) conditions in that season. Similarly, the evaporative index which is an indirect gauge of the physical properties of catchments [Jones et al., 2012] has significant associations as well as higher magnitude in the spring season. This analysis complements many studies which have linked the catchment properties at different scales to streamflow dynamics (Chiverton et al., 2015; Ann vann loon et al., 2015; Gaal et al., 2012; Ye et al., 2015). However, we do not want to stress on a single dominant factor affecting the streamflow elasticities, since there appears to be a strong interplay between elasticites and all the considered catchment properties with substantial seasonal variations."*

[Figure]

***Figure****: The linear and nonlinear association strengths as determined by Pearson and distance correlation coefficients.[Note: In the figure, we have sorted the strength of association separately for each season and the hydroclimatic variables are represented by different colors and only statistically significant [p<0.05] correlation strengths are shown here.]*

Conclusions: only point a) is a conclusion. Point b,c and d just summarize the 'observations'.

Author's reply: Thanks Dr. Westoff, We have changed the heading to summary and conclusions. Also, we have improved that section as follows:

*"(a) The proposed three parameter streamflow elasticity model can be a better model than the two parameter elasticity model as it underestimated the stream flow elasticity due to precipitation. This is because the three parameter model was able to account for the covariation of precipitation, potential evaporation and storage change.*

*(b) Seasonality plays a prominent role in streamflow elasticities with more complex behaviour in western USA basins. This complex behaviour may be linked to snow cover in the selected western basins. However, a dedicated study in this direction could further strengthen this hypothesis.*

*(c) The stream flow elasticities show significant nonlinear associations with the MOPEX catchment properties. However, we do not want to stress on any single dominant factor affecting the streamflow elasticities, since there appears to be a strong interplay between elasticites and catchment properties with substantial seasonal variations.*

*(d) We have tested our hypothesis based on the assumption of significant deep ground water losses at annual and seasonal scales. However due to shortage of Actual Evapotranspiration datasets, there may be uncertainties in the results and it can be improved by evaluating with high quality observations. This can be viewed as a hypothesis that remains to be tested using high quality climate data as and when available."*

**Technical corrections 1)**

P4, L11: To me it is not an empirical formula, but simply the definition of elasticity

Author's reply: The sentence has been changed to *"Schaake (1990) first derived the relationship between elasticity of runoff (Q) to precipitation (P) as:"*

2) For all symbols: use only one letter plus subscripts, since e.g. PET can also be interpreted as P times E times T.

Author's reply: These are conventional abbreviations used throughout the scientific literature. So, changing them might confuse the readers. However, we would first include the following statement, that "throughout the article, PET should always be interpreted as potential evapotranspiration. Similarly, DS should be interpreted as change in storage amount."

3) P7, L4: What is meant with 'irrespective of the sign'

Author's reply: We are changing the sentence to "As mentioned earlier, AIC can be used to compare the quality of a statistical models with the preferred model having the lowest absolute value"

4) P7, L22: refer to figure 5 instead of 4

Author's reply: We will make that change in the revised manuscript.

5) the paragraph "Streamflow elasticity due to Potential evapotranspiration:"starting at page 11, contains several sloppy typos. Please check carefully.

Author's reply: We will make that change in the revised manuscript.

6) add units to all axes and colour bars of the figures.

Author's reply: We will make that change in the revised manuscript.

**References:**

Berghuijs, W. R., Sivapalan, M., Woods, R. A., & Savenije, H. H. (2014). Patterns of similarity of seasonal water balances: A window into streamflow variability over a range of time scales. *Water Resources Research*, *50*(7), 5638-5661.

Chiverton, A., Hannaford, J., Holman, I., Corstanje, R., Prudhomme, C., Bloomfield, J., & Hess, T. M. (2015). Which catchment characteristics control the temporal dependence structure of daily river flows?. *Hydrological Processes*, *29*(6), 1353-1369.

Fu, G., Charles, S. P. and Chiew, F. H.: A two-parameter climate elasticity of streamflow index to assess climate change effects on annual streamflow, Water Resour. Res., 43, 2007.

Gaál, L., Szolgay, J., Kohnová, S., Parajka, J., Merz, R., Viglione, A., & Blöschl, G. (2012). Flood timescales: Understanding the interplay of climate and catchment processes through comparative hydrology. *Water Resources Research*, *48*(4).

Guo, Z., Dirmeyer, P. A., Koster, R. D., Sud, Y. C., Bonan, G., Oleson, K. W., ... & McGregor, J. L. (2006). GLACE: the global land-atmosphere coupling experiment. Part II: analysis. *Journal of Hydrometeorology*, *7*(4), 611-625.

Huntington, J. L., & Niswonger, R. G. (2012). Role of surface-water and groundwater interactions on projected summertime streamflow in snow dominated regions: An integrated modeling approach. *Water Resources Research*, *48*(11).

Jones, J. A., Creed, I. F., Hatcher, K. L., Warren, R. J., Adams, M. B., Benson, M. H., ... & Clow, D. W. (2012). Ecosystem processes and human influences regulate streamflow response to climate change at long-term ecological research sites. *BioScience*, *62*(4), 390-404.

Koster, R. D., Dirmeyer, P. A., Guo, Z., Bonan, G., Chan, E., Cox, P., ... & Liu, P. (2004). Regions of strong coupling between soil moisture and precipitation. *Science*, *305*(5687), 1138-1140.

Mei, R., & Wang, G. (2012). Summer land-atmosphere coupling strength in the United States: comparison among observations, reanalysis data, and numerical models. *Journal of Hydrometeorology*, *13*(3), 1010-1022.

Miguez-Macho, G., & Fan, Y. (2012). The role of groundwater in the Amazon water cycle: 1. Influence on seasonal streamflow, flooding and wetlands.*Journal of Geophysical Research: Atmospheres*, *117*(D15).

Van Loon, A. F., & Laaha, G. (2015). Hydrological drought severity explained by climate and catchment characteristics. *Journal of Hydrology*, *526*, 3-14.

Wang, S., Kang, S., Zhang, L., & Li, F. (2008). Modelling hydrological response to different land-use and climate change scenarios in the Zamu River basin of northwest China. *Hydrological Processes*, *22*(14), 2502-2510.

Wang, D., & Hejazi, M. (2011). Quantifying the relative contribution of the climate and direct human impacts on mean annual streamflow in the contiguous United States. *Water Resources Research*, *47*(10).

Wang, T., Istanbulluoglu, E., Lenters, J., & Scott, D. (2009). On the role of groundwater and soil texture in the regional water balance: an investigation of the Nebraska Sand Hills, USA. *Water Resources Research*, *45*(10).

Wang, D., & Hejazi, M. (2011). Quantifying the relative contribution of the climate and direct human impacts on mean annual streamflow in the contiguous United States. *Water Resources Research*, *47*(10).

Wang, X.: Advances in separating effects of climate variability and human activity on stream discharge: An overview, Adv. Water Resour., 71, 209-218, 2014.

Zhao, F., Xu, Z., Zhang, L., & Zuo, D. (2009). Streamflow response to climate variability and human activities in the upper catchment of the Yellow River Basin. *Science in China Series E: Technological Sciences*, *52*(11), 3249-3256.

Zhao, G., Hörmann, G., Fohrer, N., Zhang, Z., & Zhai, J. (2010). Streamflow trends and climate variability impacts in Poyang Lake Basin, China. *Water resources management*, *24*(4), 689-706.

---

## Author Comment (AC3) · 11 May 2016

**Response to Anonymous Reviewer**

We very much appreciate anonymous reviewer for valuable comments that helped us to improve the manuscript. Note: The text provided in italics will be incorporated in the revised draft.

**Suggestion # 1.**

*The authors used equation (4) to evaluate the impact on streamflow from precipitation, potential evapotranspiration, and storage change. In this equation, the authors estimated storage change as "(DSt - DS)/DS". I don't think that it is a good choice. According to the definition of this manuscript, DS is the long-term average of storage change, and it means that DS generally approaches zero in many basins (if there is no storage change). Therefore, it will lead to infinity for the third term on the right side of equation (4). In addition, the sign of storage change elasticity depends on the sign of DS. Consequently, we can't judge whether increasing storage leads to decreasing streamflow according to positive storage change elasticity. In that case, I suggest using storage replacing DS, or using storage change replacing "(DSt - DS)/DS".*

**Author's reply:** Yes, we do agree that at annual scale (equation 4), the DS may tent to zero. In this article, we have hypothesized that, if DS is zero, then we can always go back to equation 2 (two parameter equation), which is defined for situations were DS is zero. But, the assumption of DS =0 is not always valid. For example, in regions were the anisotropy ratio (Vertical hydraulic conductivity/Horizontal hydraulic conductivity) is not negligible, the ground water losses do occur, indicating that DS ≠0 (Wang, 2014). However, we do agree that the DS is calculated as a residual (P-Q-AET) and likely to have uncertainties due to usage of data from different sources. So, this may result in either underestimation or overestimation of storage change than it is derived in this article. Hence, until more high quality climate information is available, this can be deemed as a hypothesis that remains to be tested. However, in this study we do neglect the regions for which DS =0 at all scales. But, when it comes to seasonal scales, the DS would theoretically not be zero since, in a particular season; water balance would contain deficits or excesses depending on occurrence of rainfall events and change in temperature.

**Suggestion # 2.** *The structure of this manuscript. In Section 3, the first paragraph represents how to obtain the results. It is better to remove it into Section Methodology. Similarly, first paragraph of Section 3.5 should be removed. In P.4, the sentences from line 9-21 review the researches on the elasticity, and it is better to remove them into the Section Introduction.*

**Author's reply:** Thanks for the suggestion. These changes would be incorporated in the revised Manuscript.

Figure 8 shows that the potential evapotranspiration elasticity is larger than 0 in some basins and less than 0 in the other basins. It indicates that increasing potential evapotranspiration leads to increasing streamflow in some basins but leads to decreasing streamflow in the other basins. On the causes for the opposite impacts on streamflow, more explanations and discussions are required.

**Author's reply:**

This is a very relevant comment which requires a dedicated and separate study. However to support our findings we are including the following discussion in the revised manuscript

*"In previous studies also, certain catchments have shown positive streamflow elasticities due to potential evapotranspiration [Andréassian et al., 2015, Yang et al., 2014]. The positive PET elasticity may be caused by the local climate feedback. According to previous studies (e.g., (Koster et al. 2004; Guo et al., 2006 Mei and Wang, 2011), the central USA has strong land-atmosphere coupling strength. The PET plays an important role in the linkage of soil moisture and precipitation in the land-atmosphere interactions. Based on the positive land-atmosphere interactions, the increased soil moisture would lead to a cascading effect of increase of temperature (indirectly PET) and precipitation. The increased precipitation would therefore lead to the increase of Streamflow. In this notation, the PET has a positive relationship with precipitation, which would lead to a positive PET elasticity. The positive PET elasticity are within these hotspots in summer season".*

**Suggestion # 4.**

Figure 9 shows that the storage change elasticity is larger than 0 for many basins in spring and summer. It means that declining storage will lead to a decreasing stream- flow in those basins. At the same time, the storage change elasticity is less than 0 for other basins in spring and summer, which means declining storage resulting in increasing streamflow. The underlying mechanisms of the phenomenon should be explained and discussed.

**Author's reply:** We will include the following discussion in the revised manuscript to address your suggestion.

*"The seasonal DS elasticities indicate that ground water storage act as a natural reservoir and subsequently supply and store the streamflow during various seasons. For example, during summer when the temperatures are high and water requirement is more, ground water supplies water to the streamflow resulting in a positive elasticity in most of the MOPEX basins. Whereas, in winter and spring soil recharges itself with water indicating negative elasticity values. However, we observed that in western USA, the negative elasticity magnitude peaks during winter unlike the rest of US MOPEX basins. This may be mainly because groundwater*

*contribution to streamflow is inversely correlated to snowmelt runoff (Huntington and Niswonger, 2012). Hence, it possibly has high negative elasticity values when the snow accumulates in winter. Whereas, when the snowmelt runoff starts in the spring it starts contributing to streamflow indicating positive elasticities."*

We selected one MOPEX basin in the southern region of Florida and two basins in the state of New Mexico with the following basin ids, 94975000, 2273000 and 2296750 respectively. We investigated the summer season flows, since we suspected some anomalous behavior due to their negative elasticity values. We plot the seasonal averages of the selected time period. The streamflow and evapotranspiration are lower than rainfall amounts. The values seem normal and do not indicate an anomalous behavior. However, we do acknowledge the fact that the streamflow in those catchments is influenced by storage facilities (Wang and Hejazhi, 2012), therefore additional research is expected to address whether this is a natural behavior of the catchment.

[Figure]

**Minor comments:** 1. On the meanings of AIC and BIC, more explanations are required, i.e. why "the preferred model is the one in which the AIC value would be minimum."

**Author's reply:** We have modified the methodology section to be clearer on AIC and BIC definitions. The revised text is as follows:

*We evaluated our trivariate elasticity model (Equation 4) against the bivariate elasticity regression model (equation 2) using Akaike information criterion (AIC) (Akiake, 1973) and Bayesian Information Criterion (BIC) (Schwarz, 1978).*

*AIC is given by equation as*

$$AIC = -2\sum_{i=1}^{n} \log\{g(x_i \mid \theta_k)\} + 2k$$

*(5)*

*Where n is the number of observations; g(x) can be either equation (4) or equation (2); $\theta_k$ are the streamflow elasticities of the corresponding models and k is the number of parameters. In our context, AIC offers a relative estimate of the information lost when elasticity model is fitted to the data to represent the processes involved. As, when building any statistical model, our aim is to model the processes with minimum information loss (better goodness of fit), the preferred model is the one in which the absolute value of AIC value would be minimum. As evident from the equation (5), we can see that the first term in the equation tends to decrease with the model parameters, whereas the second term increases. Hence, AIC penalizes for the increase in number of parameters.*

*Another metric useful for calculating information loss similar to AIC is called Bayesian Information Criterion (BIC). It is computed using following equation:*

$$BIC = -2\sum_{i=1}^{n} \log\{g(x_i \mid \theta_k)\} + \log(\text{n})k$$

*(6)*

*As we can see that the BIC is similar to AIC except that the second term is multiplied by a factor of 0.5ln(n) with respect to AIC. As a result, BIC leans more towards less parameterized models. Hence, BIC should also be interpreted in a similar way as AIC. The only difference is that BIC gives more weightage to the number of parameters in a model and penalises more for the modified trivariate modeling our context. Overall, the preferred model would be the one which has both minimum AIC and BIC value.*

**Minor comments # 2**. P.2, line 5-6, Wand and Wang (2011) should be Yang and Yang (2011).
**# 3.** P.3, line 8, please check the reference Jiali et al., 2014.
**# 4.** Figure 1, the unit of the legend is missing. 5. P.2, line 19, P.7, line6, and so on, "lesser" should be "less".
**Author's reply:** The following suggestion would be implemented in the revised manuscript

**References:**

Koster, R. D., Dirmeyer, P. A., Guo, Z., Bonan, G., Chan, E., Cox, P., ... & Liu, P. (2004). Regions of strong coupling between soil moisture and precipitation. *Science*, *305*(5687), 1138-1140.

Guo, Z., Dirmeyer, P. A., Koster, R. D., Sud, Y. C., Bonan, G., Oleson, K. W., ... & McGregor, J. L. (2006). GLACE: the global land-atmosphere coupling experiment. Part II: analysis. *Journal of Hydrometeorology*, *7*(4), 611-625.

Huntington, J. L., & Niswonger, R. G. (2012). Role of surface-water and groundwater interactions on projected summertime streamflow in snow dominated regions: An integrated modeling approach. *Water Resources Research*, *48*(11).

Mei, R., & Wang, G. (2012). Summer land-atmosphere coupling strength in the United States: comparison among observations, reanalysis data, and numerical models. *Journal of Hydrometeorology*, *13*(3), 1010-1022.

Schwarz, G.: Estimating the dimension of a model, the annals of statistics, 6, 461-464, 1978.

Wang, X.: Advances in separating effects of climate variability and human activity on stream discharge: An overview, Adv. Water Resour., 71, 209-218, 2014.

---

## Author Comment (AC4) · 11 May 2016

Dear Editor:

We have addressed the reviewers suggestions and comments adequately. We will look forward to your decision on our manuscript.

Best regards Dr. Mishra Goutam Konapala